# The exocyst complex controls multiple events in the pathway of regulated exocytosis

**Sofía Suárez Freire[1,2], Sebastián Perez-Pandolfo[1,2], Sabrina Micaela Fresco[1], Julián Valinoti[1], Eleonora Sorianello[2,3], Pablo Wappner[1,2,4]\*, Mariana Melani[1,2,4]\***

[1]Fundación Instituto Leloir, Buenos Aires, Argentina; [2]Consejo Nacional De Investigaciones Científicas Y Técnicas (CONICET), Buenos Aries, Argentina; [3]Laboratorio de Regulación Hipofisaria, Instituto de Biología y Medicina Experimental (IBYME-CONICET), Buenos Aires, Argentina; [4]Departamento De Fisiología, Biología Molecular Y Celular, Facultad De Ciencias Exactas Y Naturales, Universidad De Buenos Aires, Buenos Aires, Argentina

## eLife Assessment

This study makes an **important** contribution by characterizing the role of the exocyst in secretory granule exocytosis in the *Drosophila* larval salivary gland. The results are **solid** and lead to the novel interpretation that the exocyst participates not only in exocytosis, but also in earlier steps of secretory granule biogenesis and maturation. However, the authors are urged to provide additional proof that the exocyst subunit knockdowns were effective and to acknowledge the possibility that inactivation of an essential exocytosis component could indirectly affect other parts of the secretory pathway.

**\*For correspondence:**
pwappner@leloir.org.ar (PW);
melanimari@gmail.com (MM)

**Abstract** Eukaryotic cells depend on exocytosis to direct intracellularly synthesized material toward the extracellular space or the plasma membrane, so exocytosis constitutes a basic function for cellular homeostasis and communication between cells. The secretory pathway includes biogenesis of secretory granules (SGs), their maturation and fusion with the plasma membrane (exocytosis), resulting in release of SG content to the extracellular space. The larval salivary gland of *Drosophila melanogaster* is an excellent model for studying exocytosis. This gland synthesizes mucins that are packaged in SGs that sprout from the *trans*-Golgi network and then undergo a maturation process that involves homotypic fusion, condensation, and acidification. Finally, mature SGs are directed to the apical domain of the plasma membrane with which they fuse, releasing their content into the gland lumen. The exocyst is a hetero-octameric complex that participates in tethering of vesicles to the plasma membrane during constitutive exocytosis. By precise temperature-dependent gradual activation of the Gal4-UAS expression system, we have induced different levels of silencing of exocyst complex subunits, and identified three temporally distinctive steps of the regulated exocytic pathway where the exocyst is critically required: SG biogenesis, SG maturation, and SG exocytosis. Our results shed light on previously unidentified functions of the exocyst along the exocytic pathway. We propose that the exocyst acts as a general tethering factor in various steps of this cellular process.

## Introduction

Protein secretion is a fundamental process for communication between eukaryotic cells and therefore, for organismal homeostasis, reproduction, and survival. Two types of secretory processes can be distinguished based on the rate and mode of regulation of secretory vesicle release: constitutive

secretion and regulated secretion (*Morgan, 1995*). In constitutive secretion, secretory vesicles are exocytosed as they are produced. This process takes place in all eukaryotic cells, mainly to maintain homeostasis of the plasma membrane and the extracellular matrix. In contrast, regulated secretion takes place in specialized cell types, such as endocrine and exocrine cells, as well as in cells of the immune system such as platelets and neutrophils (*Aggarwal et al., 2023*; *Ley et al., 2018*). These cell types produce specialized secretory vesicles known as secretory granules (SGs), which are released in response to specific stimuli (*Kögel and Gerdes, 2010*). SGs sprout from the *trans*-Golgi network (TGN) as immature vesicles, incompetent for fusion with the plasma membrane. Maturation of SGs involves homotypic fusion and acquisition of membrane proteins, which are required for SG delivery and fusion with the plasma membrane. Finally, stimulus-driven fusion of SGs with the plasma membrane results in cargo release to the extracellular milieu (*Omar-Hmeadi and Idevall-Hagren, 2021*; *Sugita, 2008*).

The salivary gland of *Drosophila melanogaster* larvae provides several advantages for the identification of regulators of SG biogenesis, maturation, and exocytosis (*Burgess et al., 2012*; *de la Riva-Carrasco et al., 2021*; *Neuman and Bashirullah, 2018*; *Rousso et al., 2016*). At late third larval instar, salivary glands synthesize a series of mucins, collectively known as Glue proteins, which become packed in SGs. Approximately 4 hr before the onset of pupariation, concerted exocytosis of mucin-containing SGs, followed by extrusion of the mucins out of the prepupal body are essential for gluing the puparium to the substratum (*Borne et al., 2020*). Interestingly, these three events of SG development (biogenesis, maturation, and exocytosis) occur sequentially and only once in each cell of the salivary gland, that will later on be degraded during metamorphosis (*Duan et al., 2020*; *Tracy et al., 2016*). The tagging of one of these mucins with fluorophores (Sgs3-GFP or Sgs3-dsRed), (*Biyasheva et al., 2001*) combined with the large size of salivary gland cells and their SGs, have allowed high-resolution imaging and real-time traceability of SG biogenesis, maturation, and secretion, leading to the identification and characterization of dozens of factors required along the exocytic pathway (*Burgess et al., 2011*; *Neuman et al., 2022*; *Reynolds et al., 2019*; *Torres et al., 2014*; *Tran et al., 2015*).

The exocyst is a hetero-octameric protein complex identified and initially characterized in the budding yeast, and later found to be conserved across all eukaryotic organisms (*Hsu et al., 1996*; *Novick et al., 1980*; *Novick et al., 1995*; *TerBush et al., 1996*). Yeast cells bearing mutations in exocyst subunits display intracellular accumulation of secretory vesicles and defects in exocytosis (*Govindan et al., 1995*; *TerBush et al., 1996*). Molecularly, the exocyst complex participates in vesicle tethering to the plasma membrane prior to SNARE-mediated fusion (*An et al., 2021*). Recently, it was shown that it stimulates multiple steps of exocytic SNARE complex assembly and vesicle fusion (*Lee et al., 2024*). Exocyst complex malfunction has been therefore associated with tumor growth and invasion, as well as with development of ciliopathies, among other pathological conditions (*Luo et al., 2013*; *Mavor et al., 2016*; *Thapa et al., 2012*; *Whyte and Munro, 2002*; *Wu and Guo, 2015*).

In this work, we have utilized the *Drosophila* salivary gland to carry out a methodic analysis of the requirement of each of the eight subunits of the exocyst along the regulated exocytic pathway. By inducing temperature-dependent gradual downregulation of the expression of each of the subunits, we discovered novel functions of the exocyst complex in regulated exocytosis, namely in SG biogenesis, SG maturation and homotypic fusion, as well as in SG fusion with the plasma membrane. We propose that the exocyst complex participates in all these processes as a general tethering factor.

## Results
### Characterization of SG progression during salivary gland development

Salivary glands of *D. melanogaster* larvae fulfill different functions during larval development. Mostly, they act as exocrine glands producing non-digestive enzymes during the larval feeding period, as well as mucins when the larva is about to pupariate (*Costantino et al., 2008*; *Farkaš et al., 2014*). More recently, salivary glands have been proposed to behave as endocrine organs as well, secreting a yet unidentified factor that regulates larval growth (*Li et al., 2022*). Biosynthesis of mucins produced by salivary glands, named Salivary gland secreted proteins (Sgs), begins at the second half of the third larval instar in response to an ecdysone peak (*Biyasheva et al., 2001*). After being glycosylated at the endoplasmic reticulum (ER) and the Golgi complex (GC), mucins are packed in SGs. Following subsequent developmentally controlled hormonal stimuli, SGs are massively exocytosed, releasing

the mucins to the gland lumen, and finally expelling them outside of the puparium, and gluing it to the substratum (*Biyasheva et al., 2001*). We used Sgs3-GFP or Sgs3-dsRed transgenic lines to follow this process *in vivo*. In wandering larvae, Sgs3-dsRed can be detected in salivary glands (*Figure 1A′*), while in prepupae, Sgs3-dsRed has been secreted out from the puparium, and is no longer detectable in salivary glands (*Figure 1B′*). To have a better temporal and spatial resolution of this process, we dissected and analyzed by confocal microscopy salivary glands expressing Sgs3-GFP at 4-hr intervals starting at 96 h after egg laying (AEL). Expression of Sgs3-GFP begins at 96–100 h AEL, and can be detected at the distal region of the gland (*Figure 1C*). Later on, Sgs3-GFP expression expands to more proximal cells, and at 116 h AEL, the mucin becomes detectable in the whole gland, with the exception of ductal cells which do not express Sgs3 (*Figure 1C*; *Biyasheva et al., 2001*). Thereafter, at 116–120 h AEL, in response to an ecdysone peak, SGs fuse with the apical plasma membrane, and release their content to the gland lumen (*Figure 1C*).

Detailed observation of distal cells of salivary glands revealed that SGs enlarge from 96 h AEL onwards (*Figure 1D*; *Ma and Brill, 2021b*; *Neuman et al., 2021*), a phenomenon that we quantified by measuring SG diameter (*Figure 1D* and *Table 1*). At 96–100 h AEL nascent SGs are smaller than 1 μm in diameter. Later, and up to 112 h AEL, most SGs are smaller than 3 μm in diameter, and classified them as 'Immature SGs'. From 112 h AEL onwards most SGs are bigger than 3 μm in diameter and classified as 'Mature SGs' (*Figure 1D* and *Table 1*). The maximal SG diameter that we detected was 7.13 μm at 116–120 h AEL, just before exocytosis (*Table 1*).

## The eight subunits of the exocyst complex are required for regulated exocytosis of SGs

Fluorophore-tagged Sgs3 can also be used to screen for exocytosis mutants in which Sgs3-GFP is retained inside salivary glands at the prepupal stage (*de la Riva-Carrasco et al., 2021*; *Ma et al., 2020*). Using this approach, we found that the exocyst complex is apparently required for exocytosis of SGs, since silencing of any of the eight subunits of the complex results in retention of Sgs3-GFP in salivary glands of prepupae (*Figure 2* and *Figure 3—figure supplement 1*). We used the salivary gland-specific driver *forkhead*-Gal4 (*fkh*-Gal4) to induce the expression of RNAis against each of the eight subunits of the exocyst complex in larvae that also expressed Sgs3-GFP. We analyzed Sgs3-GFP distribution in wandering larvae and prepupae (*Figure 2A, B*), and observed that whereas control prepupae were able to expel Sgs3 outside the puparium (*Figure 2A′, C*), this process was blocked in most individuals expressing RNAis against any of the exocyst subunits, with Sgs3-GFP being detected inside salivary glands of prepupae (*Figure 2B′, C* and *Figure 3—figure supplement 1*). These observations suggest that all exocyst subunits are required for exocytosis of SGs (*Table 2*).

Having established that knock-down of each of the exocyst subunits can block secretion of Sgs3, leading to retention of the mucin in the salivary glands (*Figure 2*), we investigated if exocytosis is actually impaired in these larvae. We dissected salivary glands of control and exocyst knock-down individuals at 116 h AEL, and analyzed them under the confocal microscope. Whereas control salivary gland cells were filled with mature SGs (*Figure 3A*), a more heterogeneous situation was found in salivary gland cells expressing RNAi against exocyst subunits, with some cells displaying mature SGs, and others, immature SGs, while even some displayed Sgs3-GFP in a mesh- or network-like structure (*Figure 3B*). Therefore, we investigated if this mosaic phenotypic manifestation was due to variations in cell-to-cell Gal4-UAS activation and therefore RNAi expression, being an indication of potentially different functions of the exocyst complex in the secretory pathway of salivary gland cells. We performed RNAi-mediated knock-down of each of the eight exocyst subunits at different temperatures (29, 25, 21, and 19°C) that were accurately controlled in water baths to obtain different levels of silencing and likely different phenotypic manifestations of exocyst complex downregulation. Whereas expression of a control RNAi resulted in mature SGs irrespectively of the temperature, expression of exocyst subunits RNAis at high temperatures (29°C) resulted, as a general rule, in phenotypes consistent with early arrest of the secretory pathway, since in most cells Sgs3-GFP was retained in a reticular structure or packed in immature SGs (*Figure 3C*, *Figure 3—figure supplement 1A*, and *Table 3*). By lowering the temperature of RNAi expression, and therefore moderating silencing, the most severe phenotypes (early arrest of the secretory pathway) gradually became less prominent and simultaneously, the proportion of cells displaying immature SGs, and even mature SGs, became more noticeable (*Figure 3C*, *Figure 3—figure supplement 1A*, and *Table 3*), suggesting that lower

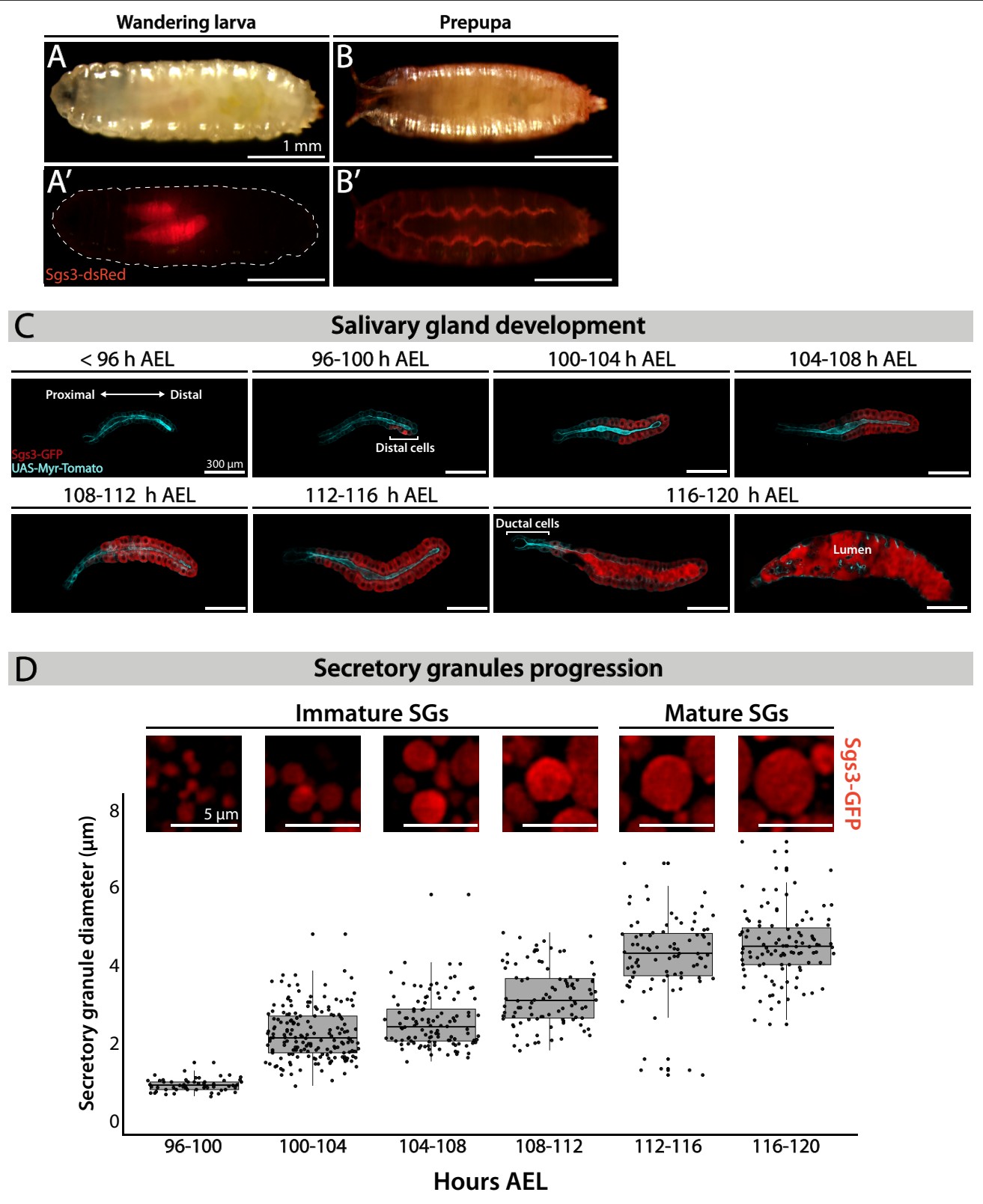

**Figure 1.** Larval salivary gland as a model for regulated exocytosis. Images of a wandering larva (**A, A'**) and a prepupa (**B, B'**) expressing Sgs3-dsRed and visualized under white light (**A, B**) or by epifluorescence (**A', B'**). Sgs3-dsRed localized in salivary glands of wandering larvae (**A'**) and outside the puparium in prepupae (**B'**). Scale bar 1 mm. (**C**) Confocal images of unfixed salivary glands dissected from larvae at the indicated time intervals (h AEL = hours after egg laying). Sgs3-GFP is shown in red and the plasma membrane labeled with myr-Tomato (*sgs3-GFP, fkh-Gal4/UAS-myr-Tomato*) is shown

*Figure 1 continued on next page*

*Figure 1 continued*

in cyan. At 96 h AEL, Sgs3 synthesis could be detected in the distal cells of the gland. Thereafter, Sgs3 expression gradually expanded proximally, and at 116 h AEL all salivary gland cells expressed Sgs3, with the exception of ductal cells. Exocytosis of secretory granules (SGs) began by this time point, and Sgs3 could be detected in the gland lumen. At 120 h AEL concerted exocytosis of SGs has ended. Scale bar 300 μm. (**D**) Confocal images of SGs labeled with Sgs3-GFP; SG diameter distribution at each time interval is displayed below. Based on its diameter, SGs are classified as immature (diameter <3 μm) or mature (diameter ≥3 μm). Only SGs from distal cells were used for diameter determination. Data points for this graph are shown in *Table 1*. For all time intervals analyzed *n* = 3, except for the 108–112 h AEL interval for which *n* = 4. '*n*' represents the number of salivary glands analyzed. Scale bar 5 μm.

The online version of this article includes the following source data for figure 1:

**Source data 1.** Raw data used to generate Figure 1D.

expression of RNAis allowed the progression of the secretory pathway. Notably, for each of the RNAis tested there was a temperature at which each of the three phenotypic outcomes could be clearly identified, although this particular temperature could be different for each RNAi, likely due to expression levels of the transgenes (*Figure 3C*, *Figure 3—figure supplement 1A*, and *Table 3*). RT-qPCR confirmed that different culturing temperatures resulted in different degrees of silencing of exocyst subunit mRNAs, as shown in *Figure 3—figure supplement 2* for *exo70* and *sec3* (*Figure 3—figure supplement 2A, B*). Also, we found that different phenotypic outcomes resulting from expression of different RNAi transgenic lines that target the same subunit (*exo70*) are due to differences in the levels of mRNA downregulation generated by each particular RNAi (*exo70* $^{RNAiV}$ or *exo70*$^{RNAiBL}$), and that there is a correlation between the level of mRNA downregulation and the strength of the phenotype observed (*Figure 3—figure supplement 2C*).

To define more precisely the role of the exocyst, and to rule out potential pleiotropic effects due to developmental defects derived from chronic exocyst downregulation, we made use of the Gal80 thermosensitive tool (*Gal80*$^{ts}$) (*Lee and Luo, 1999*). Larvae were grown at a permissive temperature (18°C) until they reached the 3$^{rd}$ instar (120 hr). In this manner, the exocyst complex could be functional until that developmental stage. Then, larvae were transferred to the restrictive temperature (29°C) and salivary glands were dissected 36 h later. We found that temporally restricted expression of *sec3*$^{RNAi}$ or *sec15*$^{RNAi}$ phenocopied unrestricted expression of the same RNAis at 29°C (*Figure 3—figure supplement 3*), indicating that the phenotypes obtained are not due to pleiotropic effects caused by developmentally unrestricted downregulation of the exocyst. Finally, the MLS phenotype generated by expression of *sec15*$^{RNAi}$ could be rescued by simultaneous expression of GFP-Sec15, supporting the notion that defective biogenesis of SGs was specifically provoked by *sec15* loss of function (*Figure 3—figure supplement 4*).

A comprehensive analysis of cell polarity, as well as a number of general markers of cellular homeostasis (*Figure 3—figure supplements 5–7*), ruled out that defects in SG biogenesis or maturation observed after knock-down of exocyst subunits could stem from potential secondary effects derived from poor cellular health, but rather reflect genuine functions of the exocyst complex in the secretory pathway. Along these lines, a recent report showed that apical polarity defects generated by loss

**Table 1.** Quantification of secretory granule (SG) diameter at the indicated time intervals after egg laying.
SGs from salivary gland distal-most cells were analyzed. Columns display: Hours AEL: hours after egg laying; *n*: number of salivary glands analyzed; number of cells analyzed; number of SGs measured; mean diameter; median diameter; minimum diameter and maximum diameter.

| Hours AEL | *n* | Cells quantified | SG quantified | Mean diameter (μm) | Median diameter (μm) | Minimum diameter (μm) | Maximum diameter (μm) |
|---|---|---|---|---|---|---|---|
| 96–100 | 3 | 7 | 57 | 0.92 | 0.91 | 0.62 | 1.49 |
| 100–104 | 3 | 18 | 173 | 2.22 | 2.11 | 0.89 | 4.76 |
| 104–108 | 3 | 12 | 112 | 2.53 | 2.53 | 1.50 | 5.78 |
| 108–112 | 4 | 13 | 96 | 3.18 | 3.18 | 1.78 | 4.81 |
| 112–116 | 3 | 10 | 86 | 4.22 | 4.22 | 1.16 | 6.59 |
| 116–120 | 3 | 15 | 107 | 4.48 | 4.45 | 2.47 | 7.13 |

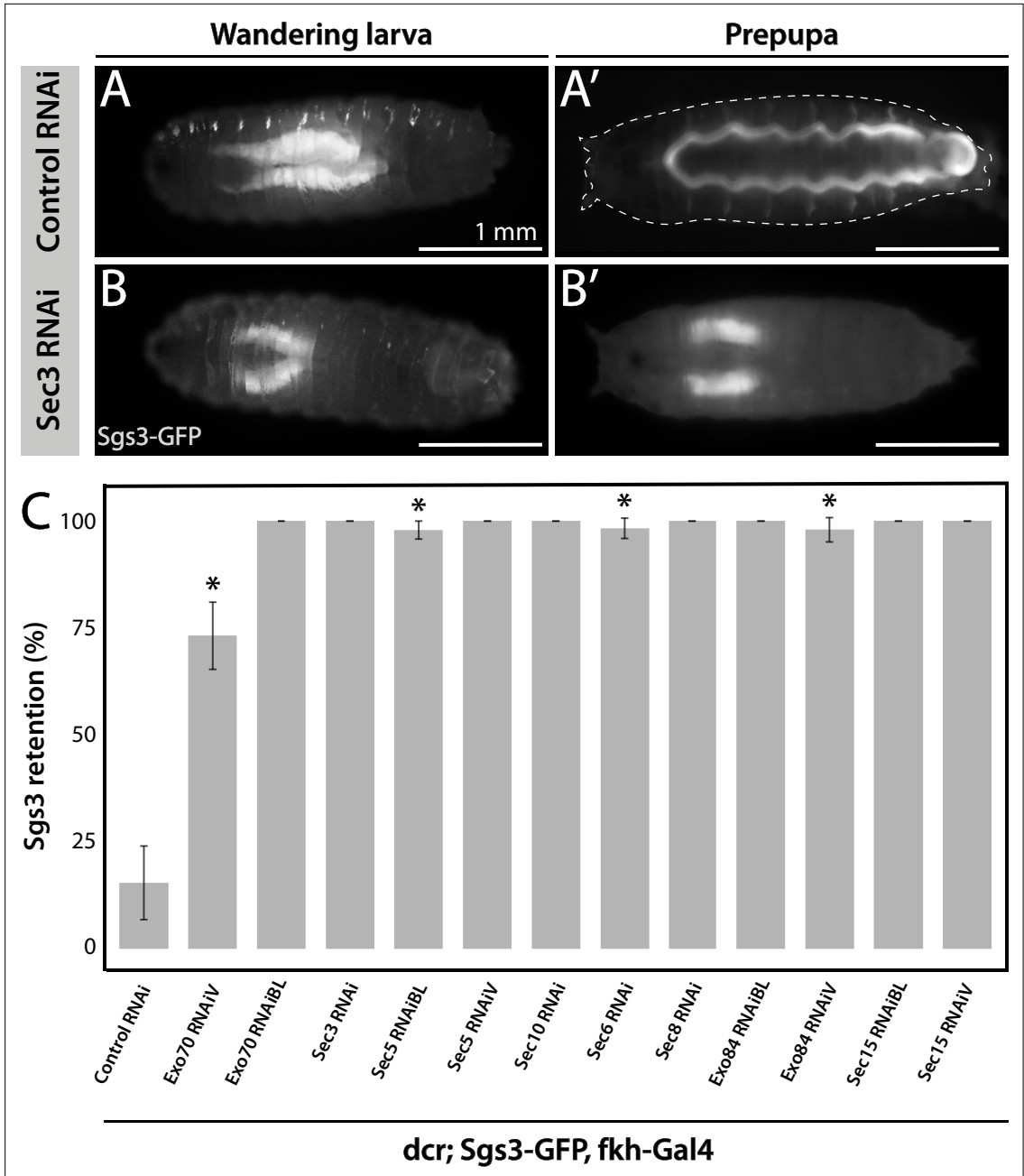

**Figure 2.** The exocyst is required for Sgs3 secretion. (**A, B**) Images of a larva and a prepupa expressing Sgs3-GFP, and visualized with epifluorescence. Sgs3-GFP was inside the salivary glands in control wandering larvae (**A**), and outside the puparium (**A'**) in control prepupae (*sgs3-GFP, fkh-Gal4/UAS-cherry^RNAi*). Expression of *sec3^RNAi* in salivary glands (*UAS-sec3^RNAi; sgs3-GFP, fkh-Gal4*) did not affect expression of Sgs3-GFP in larvae (**B**), but blocked Sgs3-GFP release outside the puparium, so the protein was retained inside the salivary glands (**B'**). This phenotypic manifestation is referred as 'retention phenotype'. Scale bar 1 mm. (**C**) Quantification of the penetrance of the retention phenotype in prepupae expressing the indicated RNAis. RNAis were expressed using *fkh-Gal4* and larvae were cultured at 29°C. All RNAis tested against exocyst complex subunits displayed a retention phenotype, with a penetrance significantly different from the control RNAi (*UAS-cherry^RNAi*) according to Likelihood ratio test followed by Tukey's test (* = p-value <0.05). *cherry^RNAi* n = 7; *exo70^RNAiV* n = 11; *exo70^RNAiBL* n = 8; *sec3^RNAi* n = 6; *sec5^RNAiBL* n = 17; *sec5^RNAiV* n = 6; *sec10^RNAi* n = 9; *sec6^RNAi* n = 11; *sec8^RNAi* n = 5; *exo84^RNAiBL* n = 5; *exo84^RNAiV*, n = 6; *sec15^RNAiBL* n = 3; *sec15^RNAiV* n = 4. 'n' represents the number of vials containing 20–30 prepupae per vial. For exocyst subunits with more than one RNAi line available, 'BL' indicates a Bloomington Stock Center allele and 'V' a Vienna Stock Center allele (see *Table 2* for stock numbers).

The online version of this article includes the following source data for figure 2:

**Source data 1.** Raw data used to generate *Figure 2C*.

**Table 2.** List of the *Drosophila* lines utilized in this work.
Stock number and repository center from where each line was obtained are indicated.

| Line | Number | Stock center |
| --- | --- | --- |
| Sgs3-GFP | 5885 | Bloomington |
| Sgs3-dsRed | - | A.J. Andres' Lab |
| YFP-Rab11 | 62549 | Bloomington |
| YFP-Rab1 | 62539 | Bloomington |
| UAS-dicer2 | 24650 | Bloomington |
| UAS-White-RNAi | 33613 | Bloomington |
| UAS-Cherry-RNAi | 35785 | Bloomington |
| UAS-Rab11-RNAi | 27730 | Bloomington |
| UAS-Sec3-RNAi | 35806 | Vienna |
| UAS-Sec5-RNAi | 27526 | Bloomington |
| UAS-Sec5-RNAi | 28873 | Vienna |
| UAS-Sec6-RNAi | 27314 | Bloomington |
| UAS-Sec8-RNAi | 45032 | Vienna |
| UAS-Sec10-RNAi | 27483 | Bloomington |
| UAS-Sec15-RNAi | 27499 | Bloomington |
| UAS-Sec15-RNAi | 35161 | Vienna |
| UAS-Exo84-RNAi | 108650 | Vienna |
| UAS-Exo84-RNAi | 28712 | Bloomington |
| UAS-Exo70-RNAi | 27867 | Vienna |
| UAS-Exo70-RNAi | 28041 | Bloomington |
| UAS-PLCγ-PH-EGFP | 58362 | Bloomington |
| UAS-Synaptotagmin1-GFP | 6925 | Bloomington |
| UAS-Sec8 | 9556 | Bloomington |
| UAS-CD63-GFP | 91390 | Bloomington |
| UAS-Myr-Tomato | 32221 | Bloomington |
| UAS-GFP-KDEL | 9898 | Bloomington |
| UAS-Bip-SfGFP-HDEL | 64749 | Bloomington |
| UAS-RFP-Golgi | 30908 | Bloomington |
| UAS-GRASP65-GFP | 8508 | Bloomington |
| UAS-EGFP | 5430 | Bloomington |
| UAS-GFP-Sec15 | 39685 | Bloomington |
| UAS-YFP-Rab11$^{CA}$ | 9791 | Bloomington |
| tubP-GAL80[ts] | 7017 | Bloomington |
| UAS-mCD8-mCherry | 27391 | Bloomington |
| UAS-mito-GFP | 8443 | Bloomington |
| UASp-GFP-mCherry-Atg8a | 37749 | Bloomington |

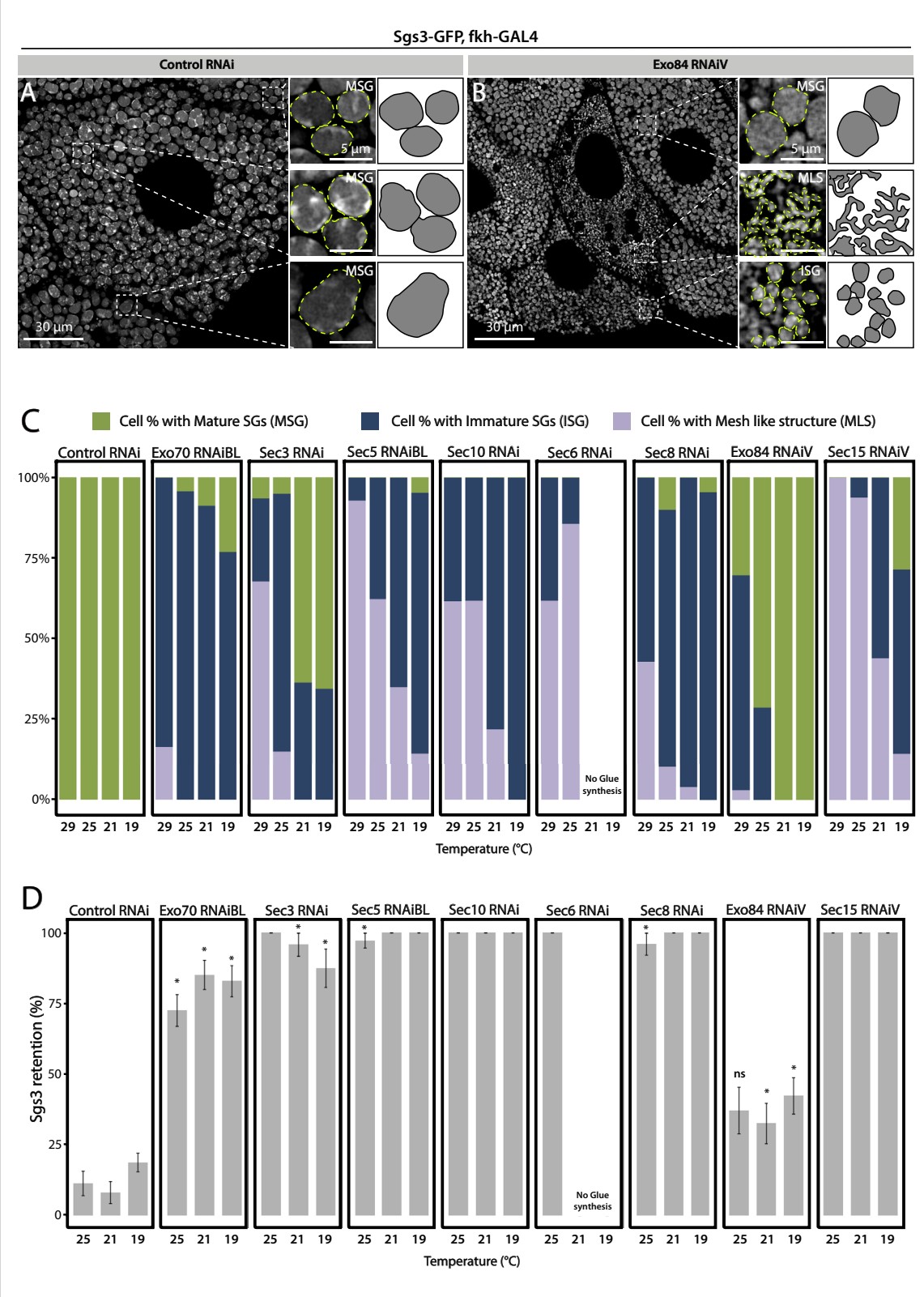

**Figure 3.** Phenotypic manifestations of exocyst subunits silencing. At the end of larval development (116–120 h AEL) salivary gland cells of control individuals (*sgs3-GFP*, *fkh-Gal4/UAS-cherry^RNAi*) (**A**) were filled with mature secretory granules (SGs) (insets). In cells expressing *exo84^RNAiV* (*UAS-exo84^RNAiV*; *sgs3-GFP*, *fkh-Gal4*) (**B**), three different phenotypes could be visualized in a single salivary gland: cells with mature SGs (MSG), cells with immature SGs (ISG), and cells with no SG, in which Sgs3 was retained in a mesh (MLS). Scale bar 30 μm in main panels, and 5 μm in insets. For didactic purposes, MSG,

*Figure 3 continued on next page*

eLife Research article

Cell Biology | Developmental Biology

*Figure 3 continued*

ISG, and MLS were drawn on the right, next to the corresponding inset. (**C**) Quantification of the penetrance of each of the three phenotypes observed upon downregulation of each of the exocyst subunits. Larvae were grown at four different temperatures (29, 25, 21, or 19°C) to achieve different levels of RNAi expression. '*n*' = number of salivary glands analyzed; *control*$^{RNAi}$ (*cherry*$^{RNAi}$) *n* = 4 (29°C), *n* = 5 (25, 21, and 19°C); *exo70*$^{RNAiBL}$ *n* = 11 (29°C), *n* = 7 (25 and 21°C), *n* = 6 (19°C); *sec3*$^{RNAi}$ *n* = 7 (29 and 25°C), *n* = 4 (21°C), *n* = 9 (19°C); *sec5*$^{RNAiBL}$ *n* = 4 (29°C), *n* = 12 (25°C), *n* = 9 (21°C), *n* = 6 (19°C); *sec10*$^{RNAi}$ *n* = 8 (29°C), *n* = 6 (25 and 19°C), *n* = 7 (21°C); *sec6*$^{RNAi}$ *n* = 9 (29°C), *n* = 4 (25°C); *sec8*$^{RNAi}$ *n* = 8 (29°C), *n* = 6 (25 and 19°C), *n* = 7 (21°C); *exo84*$^{RNAiV}$ *n* = 8 (29°C), *n* = 4 (25 and 19°C), *n* = 5 (21°C); *sec15*$^{RNAiV}$ *n* = 4 (29 and 25°C), *n* = 6 (21°C), *n* = 5 (19°C). Raw data used to generate this graph is shown in *Table 3*. (**D**) Quantification of the penetrance of the Sgs3-GFP retention phenotype in salivary glands of prepupae of the indicated genotypes. Only a few control individuals (*sgs3-GFP, fkh-Gal4/UAS-cherry*$^{RNAi}$) displayed the retention phenotype. Downregulation of exocyst subunits provoked significant retention of Sgs3 inside the salivary glands irrespective to the temperature (25, 21, or 19°C). Expression of *sec6*$^{RNAi}$ at 21 or 19°C resulted in no synthesis of Sgs3-GFP or Sgs3-dsRed, so the distribution of phenotypes was not assessed for this genotype at these temperatures. RNAis were expressed with *fkh*-Gal4. '*n*' = number of vials with 20–30 larvae. *control*$^{RNAi}$ (*cherry*$^{RNAi}$) *n* = 7 (25°C), *n* = 9 (21°C), *n* = 19 (19°C); *exo70*$^{RNAiBL}$ *n* = 13 (25°C), *n* = 9 (21°C), *n* = 22 (19°C); *sec3*$^{RNAi}$ *n* = 9 (25°C), *n* = 6 (21 and 19°C); *sec5*$^{RNAiBL}$ *n* = 9 (25°C), *n* = 7 (21°C), *n* = 12 (19°C); *sec10*$^{RNAi}$ *n* = 10 (25°C), *n* = 5 (21°C), *n* = 4 (19°C); *sec6*$^{RNAi}$ *n* = 10 (25°C); *sec8*$^{RNAi}$ *n* = 5 (25°C), *n* = 8 (21°C), *n* = 22 (19°C); *exo84*$^{RNAiV}$ *n* = 8 (25 and 19°C), *n* = 6 (21°C); *sec15*$^{RNAiV}$ *n* = 5 (25°C), *n* = 9 (21°C), *n* = 8 (19°C). Statistical analysis was performed using a Likelihood ratio test followed by Tuckey's test (*p-value <0.05). For those genotypes with 100% penetrance no statistical analysis was performed due to the lack of standard error. ns = not significant. Comparisons were made between RNAis for each of the temperatures analyzed.

The online version of this article includes the following source data and figure supplement(s) for figure 3:

**Source data 1.** Raw data used to generate *Figure 3C*.

**Source data 2.** Raw data used to generate *Figure 3D*.

**Figure supplement 1.** Penetrance of phenotypes observed after silencing subunits of the exocyst.

**Figure supplement 1—source data 1.** Raw data used to generate *Figure 3—figure supplement 1*.

**Figure supplement 2.** Remaining mRNA levels after RNAi-mediated knock-down of exocyst subunits correlate with the observed phenotypes.

**Figure supplement 3.** Chronic or acute knock-down of exocyst subunits generate comparable phenotypes.

**Figure supplement 3—source data 1.** Raw data used to generate *Figure 3—figure supplement 3*.

**Figure supplement 4.** Expression of GFP-Sec15 can rescue secretory granule (SG) maturation after *sec15* knock-down.

**Figure supplement 4—source data 1.** Raw data used to generate *Figure 3—figure supplement 4*.

**Figure supplement 5.** Exocyst down-regulation does not affect general cellular health or homeostasis.

**Figure supplement 6.** Exocyst down-regulation does not affect nuclei size or autophagy.

**Figure supplement 6—source data 1.** Raw data used to generate *Figure 3—figure supplement 6*.

**Figure supplement 7.** Exocyst down-regulation does not affect apical polarity markers.

**Figure supplement 7—source data 1.** Raw data used to generate *Figure 3—figure supplement 7*.

of the polarity protein Crumbs, do not affect or interfere with SG exocytosis (*Lattner et al., 2019*), further supporting the notion that there are parallel pathways controlling cell polarity and SG biogenesis, maturation and exocytosis.

The fact that the three phenotypic outcomes (1. Sgs3-GFP retained in a mesh; 2. immature SGs; and 3. mature-not exocytosed SGs) could be retrieved by appropriately silencing any of the eight subunits lead us to propose that the holocomplex, and not subcomplexes or individual subunits, function several times along the secretory pathway, and that each of these activities require different amounts of the exocyst complex. Importantly, irrespectively of the temperature of expression of RNAis, retention of SGs in salivary gland cells was always significantly higher in exocyst knock-down individuals as compared to controls (*Figure 3D* and *Figure 3—figure supplement 1B*), indicating that ultimately, the exocyst is required for SG exocytosis.

## The exocyst complex is required for SG biogenesis

We decided to characterize each of the three phenotypic manifestations of exocyst loss of function in more detail. The early-most manifestation of the requirement of the exocyst in the secretory pathway was the reticular or mesh-like phenotype obtained by strong silencing (29°C) of any of the subunits (*Figure 4A*). This phenotype was reminiscent of mutants in which Sgs3 is retained at ER exit sites or at ER–Golgi complex (GC) contact sites (*Burgess et al., 2011*; *Reynolds et al., 2019*), suggesting that knock-down of the exocyst may provoke Sgs3 retention at the ER or GC, blocking SG biogenesis. Indeed, we found that RNAi-mediated silencing of *sec15,sec3* or *sec10* provoked Sgs3 retention in

**Table 3.** Raw data for experiments of *Figure 3C* and *Figure 3—figure supplement 1A*.

Figure 3C

| Genotype | Temperature (°C) | Number of glands analyzed | Number of distal cells analyzed | Phenotype | % of phenotype | Standard deviation |
|---|---|---|---|---|---|---|
| | | | | Mesh-like structure | 0 | 0 |
| | | | | SG immature | 0 | 0 |
| | 29 | 4 | 12 | SG mature | 100 | 0 |
| | | | | Mesh-like structure | 0 | 0 |
| | | | | SG immature | 0 | 0 |
| | 25 | 5 | 13 | SG mature | 100 | 0 |
| | | | | Mesh-like structure | 0 | 0 |
| | | | | SG immature | 0 | 0 |
| | 21 | 5 | 17 | SG mature | 100 | 0 |
| | | | | Mesh-like structure | 0 | 0 |
| | | | | SG immature | 0 | 0 |
| Control RNAi | 19 | 5 | 15 | SG mature | 100 | 0 |
| | | | | Mesh-like structure | 12.73 | 28.67 |
| | | | | SG immature | 87.27 | 28.67 |
| | 29 | 11 | 43 | SG mature | 0 | 0 |
| | | | | Mesh-like structure | 0 | 0 |
| | | | | SG immature | 95.24 | 12.6 |
| | 25 | 5 | 23 | SG mature | 4.76 | 12.6 |
| | | | | Mesh-like structure | 0 | 0 |
| | | | | SG immature | 90.48 | 25.2 |
| | 21 | 7 | 23 | SG mature | 9.52 | 25.2 |
| | | | | Mesh-like structure | 0 | 0 |
| | | | | SG immature | 0 | 0 |
| Exo70 RNAi BL | 19 | 4 | 15 | SG mature | 100 | 0 |
| | | | | Mesh-like structure | 69.52 | 42.84 |
| | | | | SG immature | 24.05 | 32.37 |
| | 29 | 7 | 31 | SG mature | 6.43 | 11.07 |
| | | | | Mesh-like structure | 16.67 | 28.87 |
| | | | | SG immature | 78.57 | 28.41 |
| | 25 | 7 | 20 | SG mature | 4.76 | 12.6 |
| | | | | Mesh-like structure | 0 | 0 |
| | | | | SG immature | 27.08 | 35.6 |
| | 21 | 4 | 11 | SG mature | 72.92 | 35.6 |
| | | | | Mesh-like structure | 0 | 0 |
| | | | | SG immature | 33.52 | 33.75 |
| Sec3 RNAi | 19 | 9 | 32 | SG mature | 66.48 | 33.75 |

*Table 3 continued on next page*

*Table 3 continued*

**Figure 3C**

| | | | | | | |
|---|---|---|---|---|---|---|
| | | | | Mesh-like structure | 96.43 | 7.14 |
| | | | | SG immature | 3.57 | 7.14 |
| | 29 | 4 | 14 | SG mature | 0 | 0 |
| | | | | Mesh-like structure | 62.5 | 38.35 |
| | | | | SG immature | 37.5 | 38.35 |
| | 25 | 12 | 45 | SG mature | 0 | 0 |
| | | | | Mesh-like structure | 27.78 | 44.1 |
| | | | | SG immature | 72.22 | 44.1 |
| | 21 | 9 | 20 | SG mature | 0 | 0 |
| | | | | Mesh-like structure | 12.22 | 19.05 |
| | | | | SG immature | 84.44 | 18.22 |
| Sec5 RNAi BL | 19 | 6 | 21 | SG mature | 3.33 | 8.16 |
| | | | | Mesh-like structure | 77.78 | 44.1 |
| | | | | SG immature | 22.22 | 44.1 |
| | 29 | 9 | 26 | SG mature | 0 | 0 |
| | | | | Mesh-like structure | 59.17 | 46.95 |
| | | | | SG immature | 40.83 | 46.95 |
| | 25 | 6 | 34 | SG mature | 0 | 0 |
| | | | | Mesh-like structure | 20.24 | 34.65 |
| | | | | SG immature | 79.76 | 34.65 |
| | 21 | 7 | 23 | SG mature | 0 | 0 |
| | | | | Mesh-like structure | 0 | 0 |
| | | | | SG immature | 100 | 0 |
| Sec10 RNAi | 19 | 6 | 20 | SG mature | 0 | 0 |
| | | | | Mesh-like structure | 66.67 | 50 |
| | | | | SG immature | 33.33 | 50 |
| | 29 | 9 | 34 | SG mature | 0 | 0 |
| | | | | Mesh-like structure | 75 | 50 |
| | | | | SG immature | 25 | 50 |
| Sec6 RNAi | 25 | 4 | 7 | SG mature | 0 | 0 |

*Table 3 continued on next page*

*Table 3 continued*

**Figure 3C**

| | | | | | | |
|---|---|---|---|---|---|---|
| | | | | Mesh-like structure | 41.16 | 35.88 |
| | | | | SG immature | 58.84 | 35.88 |
| | 29 | 8 | 37 | SG mature | 0 | 0 |
| | | | | Mesh-like structure | 8.33 | 20.41 |
| | | | | SG immature | 80.56 | 30.58 |
| | 25 | 6 | 20 | SG mature | 11.11 | 27.22 |
| | | | | Mesh-like structure | 3.57 | 9.45 |
| | | | | SG immature | 96.43 | 9.45 |
| | 21 | 7 | 26 | SG mature | 0 | 0 |
| | | | | Mesh-like structure | 0 | 0 |
| | | | | SG immature | 96.67 | 8.16 |
| Sec8 RNAi | 19 | 6 | 22 | SG mature | 3.33 | 8.16 |
| | | | | Mesh-like structure | 2.08 | 5.89 |
| | | | | SG immature | 60.42 | 32.49 |
| | 29 | 8 | 33 | SG mature | 37.5 | 33.46 |
| | | | | Mesh-like structure | 0 | 0 |
| | | | | SG immature | 27.92 | 28 |
| | 25 | 4 | 14 | SG mature | 72.08 | 28 |
| | | | | Mesh-like structure | 0 | 0 |
| | | | | SG immature | 0 | 0 |
| | 21 | 5 | 15 | SG mature | 100 | 0 |
| | | | | Mesh-like structure | 0 | 0 |
| | | | | SG immature | 0 | 0 |
| Exo84 RNAi V | 19 | 4 | 8 | SG mature | 100 | 0 |
| | | | | Mesh-like structure | 100 | 0 |
| | | | | SG immature | 0 | 0 |
| | 29 | 4 | 17 | SG mature | 0 | 0 |
| | | | | Mesh-like structure | 87.5 | 25 |
| | | | | SG immature | 12.5 | 25 |
| | 25 | 4 | 16 | SG mature | 0 | 0 |
| | | | | Mesh-like structure | 43.33 | 46.33 |
| | | | | SG immature | 56.67 | 46.33 |
| | 21 | 6 | 25 | SG mature | 0 | 0 |
| | | | | Mesh-like structure | 16.67 | 23.57 |
| | | | | SG immature | 53.33 | 36.13 |
| Sec15 RNAi V | 19 | 5 | 14 | SG mature | 30 | 44.72 |

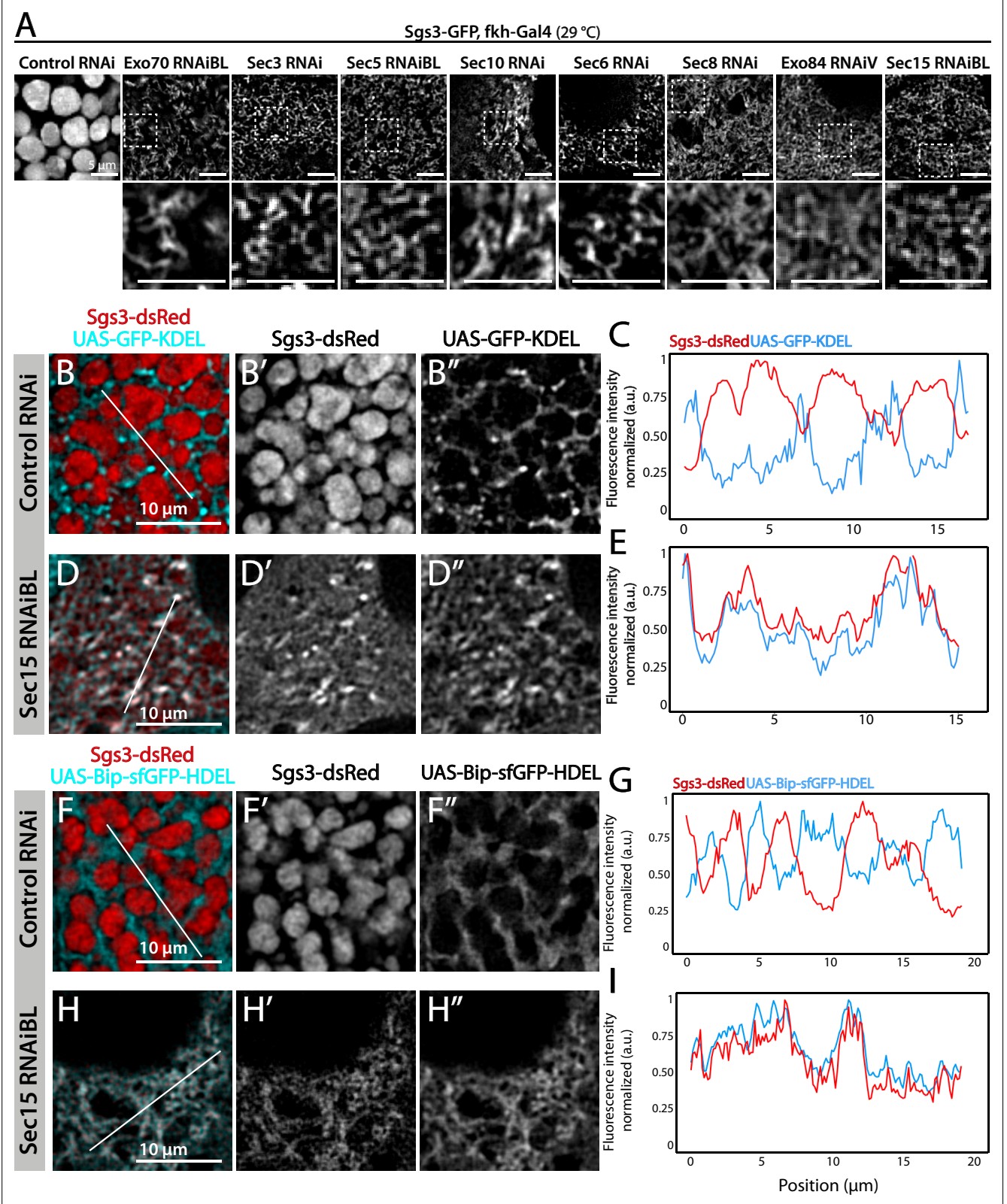

**Figure 4.** The exocyst is required for Sgs3-GFP exit from endoplasmic reticulum (ER) and secretory granule (SG) biogenesis. Confocal images of unfixed salivary glands of the indicated genotypes. Larvae were grown at 29°C to achieve maximal activation of the Gal4-UAS system, and maximal downregulation of exocyst subunits. (**A**) In control salivary glands (*sgs3-GFP*, *fkh-Gal4/UAS-cherry^RNAi^*) 116–120 AEL, Sgs3-GFP was packed in mature SGs. Salivary glands expressing RNAis against each of the exocyst subunits, where Sgs3-GFP exhibited a reticular distribution are shown. SGs were

*Figure 4 continued on next page*

*Figure 4 continued*

not formed in these cells. Scale bar 5 µm. The ER, labeled with GFP-KDEL (*UAS-GFP-KDEL*), (**B, D**) or Bip-sfGFP-HDEL (*UAS-Bip-sfGFP-HDEL*), (**F, H**) distributed in between SGs in control salivary glands (**B, F**); in *sec15^RNAi^* salivary glands (*fkh-Gal4/UAS-sec15^RNAiBL^*) SGs did not form and the Sgs3-dsRed signal overlapped with the ER markers (**D, H**). (**C, E, G, I**) Two-dimensional line scans of fluorescence intensity across the white lines in panels B, D, F, and H of Sgs3-dsRed and the ER markers. In all cases, transgenes were expressed using *fkh*-Gal4. Scale bar 10 µm.

The online version of this article includes the following source data and figure supplement(s) for figure 4:

**Source data 1.** Raw data used to generate *Figure 4C*.

**Figure supplement 1.** The exocyst is required for Sgs3-GFP exit from endoplasmic reticulum (ER) and secretory granule (SG) biogenesis.

**Figure supplement 1—source data 1.** Raw data used to generate *Figure 4—figure supplement 1*.

**Figure supplement 2.** Quantification of colocalization between endoplasmic reticulum (ER) markers and Sgs3.

**Figure supplement 2—source data 1.** Raw data used to generate *Figure 4—figure supplement 2B*.

the ER, since it colocalized with ER markers, as indicated by Pearson's coefficient, whereas in control larvae Sgs3-GFP was inside SGs, at a comparable developmental stage (*Figure 4B–G* and *Figure 4— figure supplement 1*; *Figure 4—figure supplement 2*).

We analyzed in detail GFP-Sec15 subcellular localization, and found that it associated closely with the *trans*-Golgi marker RFP-Golgi, but not with the ER marker KDEL-RFP in salivary glands just prior to SG biogenesis (*Figure 5* and *Videos 1–3*). This association was lost after the onset SG biogenesis, suggesting that the exocyst associates with the GC at the specific developmental stage when Sgs3 transits through that organelle (*Figure 5B–E*). Three-dimensional reconstruction of Sec15-Golgi *foci* confirmed the spatial association with the exocyst (*Figure 5E* and *Videos 2 and 3*). In line with this, we found that exocyst silencing under conditions that block Sgs3-GFP in the ER (29°C), both *cis* and *trans*-CG structures were severely affected (*Figure 6* and *Figure 6—figure supplement 1*). This suggests that at this stage of salivary gland development the exocyst complex localizes at the GC, where it is required to maintain *cis*- and *trans*-GC morphology, thereby allowing the correct transport of Sgs3 from the ER to the GC (*Figure 6E, F*).

## Role of the exocyst in SG maturation: Homotypic fusion

Given that appropriate silencing conditions of any of the exocyst subunits can result in accumulation of immature SGs (*Figure 7A*), we set out to investigate a potential role of the exocyst in SG maturation. SG maturation is a multifaceted process that involves, among other events, homotypic fusion between immature SGs (*Du et al., 2016*; *Neuman and Bashirullah, 2018*). We found that GFP-Sec15 often localized in discrete *foci* in between immature SGs (75% of total GFP-Sec15), but this localization drops dramatically when SGs have undergone maturation (15%) (*Figure 7B, C* and *Videos 4–5*). This transient localization of GFP-Sec15 supports the notion of a role of the exocyst in SG homofusion, which was confirmed by live imaging of salivary glands *ex vivo* (*Figure 7D* and *Video 6*). In support to this notion, when GFP-Sec15 was overexpressed at 25°C, unusually large SGs of up to 20 µm in diameter could be detected, a size never observed in control salivary glands overexpressing GFP alone (*Figure 7E–H*), indicating that overexpression of the Sec15 subunit alone is sufficient to induce homotypic fusion between SGs, which is in agreement with reports that indicate that Sec15 functions as a seed for exocyst complex assembly (*Escrevente et al., 2021*; *Guo et al., 1999*). Sec15 overexpression is expected to induce the formation of exocyst holocomplexes, provoking excessive homofusion among SGs. In contrast, overexpression of Sec8, which is not expected to induce the formation of the whole complex, did not have an effect on homotypic fusion of SGs (*Figure 7F, H*).

The characteristic localization of GFP-Sec15 *foci* in between adjacent immature SGs, the fact that Sec15 overexpression results in oversized SGs, and the observation that downregulation of any subunit of the exocyst complex at appropriate levels results in accumulation of immature SGs, weigh in favor of the notion that the exocyst plays a critical role in SG homotypic fusion.

## Role of the exocyst in SG maturation: Acquisition of membrane proteins

Besides homotypic fusion, maturation of SGs involves the incorporation of specific proteins that are required for homotypic fusion, apical navigation, or fusion with the apical plasma membrane. The mechanisms by which these maturation factors associate with SGs are not well understood. One

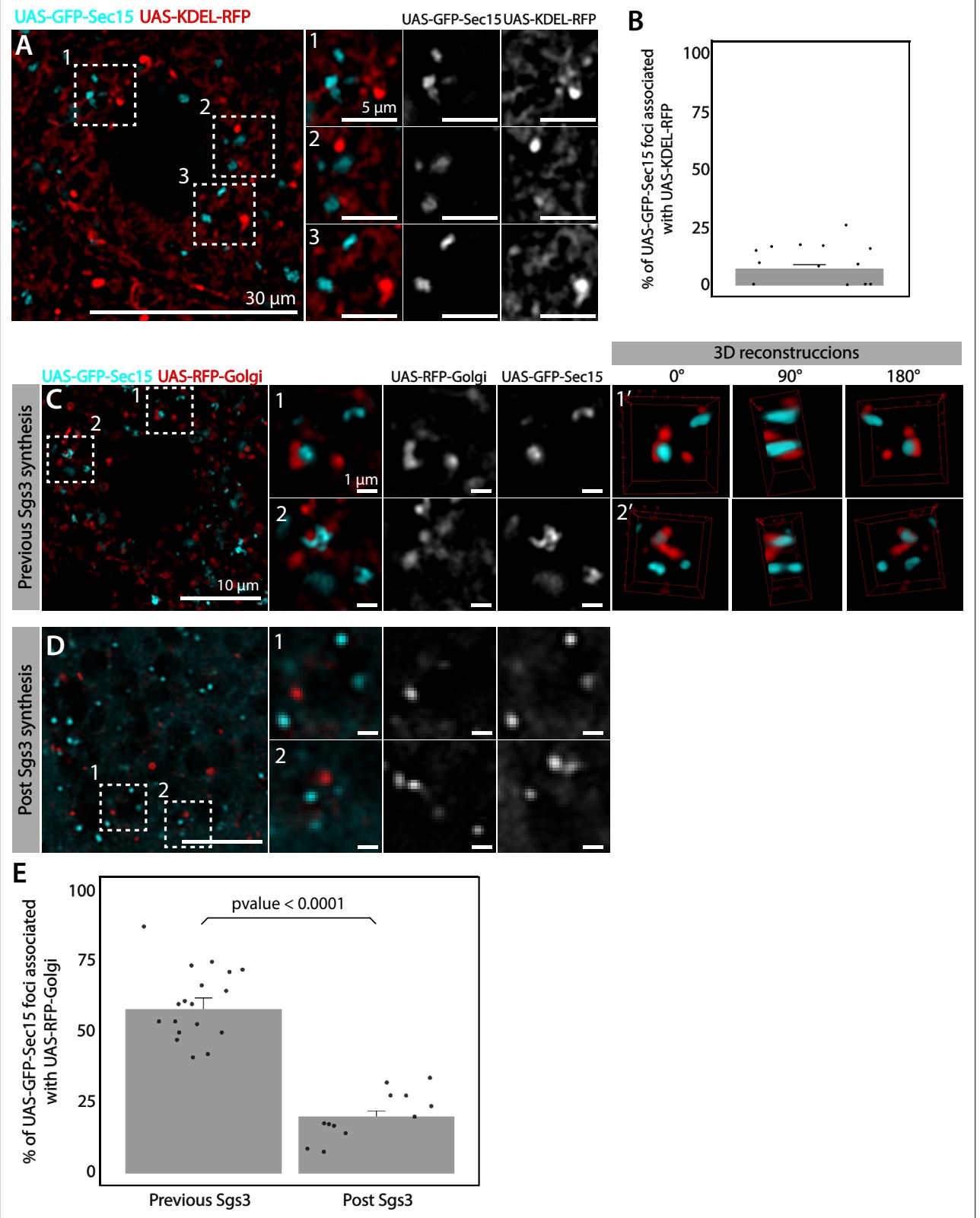

**Figure 5.** The exocyst localizes at the Golgi complex before Sgs3 synthesis. Confocal images of unfixed salivary glands expressing GFP-Sec15 and (**A**) the endoplasmic reticulum (ER) marker KDEL-RFP or (**C**) the *trans*-Golgi marker RFP-Golgi. Sec15 *foci* did not localize at or associate with the ER (**A, B**). (**C**) Sec15 *foci* were found in close association with *trans*-Golgi complex *cisternae*. Examples of association events are shown in insets 1 and 2, including different angles of three-dimensional reconstruction stacks. (**D**) Sec15 *foci* and *trans*-Golgi complex association was lost after Sgs3 synthesis

*Figure 5 continued on next page*

*Figure 5 continued*

has begun. (**E**) Quantification of Sec15 *foci*-Golgi complex association events before and after the onset of Sgs3 synthesis (Wald test, p-value <0.05). '*n*' represents the number of salivary glands analyzed, *n* = 4.

The online version of this article includes the following source data for figure 5:

**Source data 1.** Raw data used to generate *Figure 5B and E*.

of such proteins is the calcium sensor Synaptotagmin-1 (Syt-1), which localized at the basolateral membrane of salivary gland cells before SG biogenesis (96 h AEL) (*Figure 8—figure supplement 1A*), and later became detectable on the membrane of nascent SGs (diameter <1 μm), prior to homotypic fusion (*Figure 8A* and *Figure 8—figure supplement 1B*). As SGs mature, the presence of Syt-1 on SGs became more prominent, with a sharp increase after SG homofusion (*Figure 8—figure supplement 1C, D*), suggesting that recruitment of Syt-1 to SGs continued after homotypic fusion had occurred (*Figure 8M*). Downregulation of the exocyst subunits Sec5 or Sec3 to levels that generate immature SGs significantly reduced the presence of Syt-1 on their membranes, as compared to immature SGs of control salivary glands (*Figure 8A–C* and *Figure 8—figure supplement 2A–C*). Interestingly, weaker knock-down, which allowed the formation of mature SGs, improved but not completely restored Syt-1 recruitment, compared to control salivary cells (*Figure 8—figure supplement 2D–F*), indicating that Syt-1 is recruited to mature SGs in an exocyst-dependent manner (*Figure 8M*).

CD63, the *Drosophila* homolog of mammalian Tsp29Fa, is another protein required for SG maturation. CD63 localizes at the apical plasma membrane before SG formation, and reaches the membrane of SGs through endosomal retrograde trafficking (*Figure 8—figure supplement 1E*; *Ma et al., 2020*). We investigated if the exocyst participates in recruitment of CD63 to SGs, and found that CD63 could be readily detected at the membrane of 1, 3, or 5 μm SGs (*Figure 8D* and *Figure 8—figure supplement 1F–H*), while upon Sec5 or Sec3 downregulation CD63 was significantly reduced on immature SGs (*Figure 8D–F* and *Figure 8—figure supplement 2G–I*). Interestingly, milder reduction of Sec3 expression, under conditions that allow the formation of mature SGs, did not affect CD63 recruitment to SGs (*Figure 8—figure supplement 2J–L*), indicating that CD63 is recruited to SGs before homotypic fusion in an exocyst-dependent manner and that, unlike to Syt-1, recruitment ceases after homotypic fusion has occurred (*Figure 8M*).

The small GTPases Rab11 and Rab1 were reported to be required for SG maturation. Whereas Rab11 associates to SGs independently of their size, Rab1 was found transiently on immature SGs only (*Ma and Brill, 2021a*; *Neuman et al., 2021*), and moreover, Rab1 recruitment to SGs depends on Rab11 (*Neuman et al., 2021*). Given that Rab11 is a stable component of SGs and Rab1 is not, we

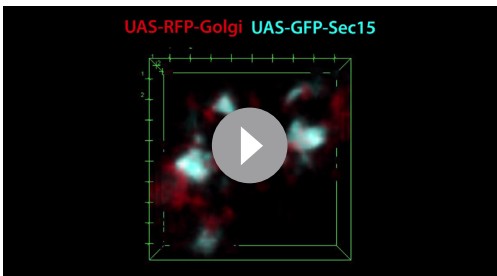

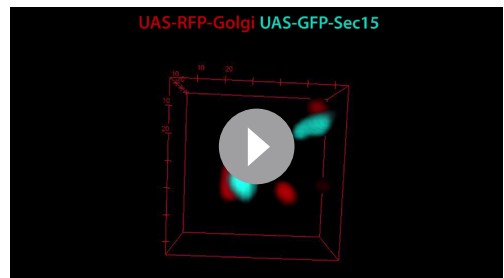

**Video 1.** Real-time imaging showing association between the exocyst and the *trans*-Golgi network. A *Drosophila* early third instar larva salivary gland (~168 hr at 18°C equivalent to ~72–96 hr at 25°C) expressing the *trans*-Golgi marker RFP-Golgi (red) and the exocyst maker GFP-Sec15 (cyan (*UAS-GFP-sec15/UAS-RFP-Golgi; fkh-Gal4*)) was imaged. Three-dimensional reconstruction of six slides with 0.44 μm spacing was generated using ImageJ. Frame acquisition time was 2.25 s. A total of 78 frames per channel were acquired. Total video time is 77 s. Crop size is 8 μm.

https://elifesciences.org/articles/92404/figures#video1

**Video 2.** Three-dimensional reconstruction showing association between the exocyst and the *trans*-Golgi network. A salivary gland from an early third instar larva (~168 hr at 18°C equivalent to ~72–96 hr at 25°C) was imaged. The *trans*-Golgi marker RFP-Golgi (red) and the exocyst maker GFP-Sec15 (cyan) were in close association (*UAS-GFP-sec15/UAS-RFP-Golgi; fkh-Gal4*). Three-dimensional reconstruction of 33 slides with 0.1 μm spacing was generated using ImageJ. Crop size is 5.57 μm. Corresponds to images shown in *Figure 5C-1'*.

https://elifesciences.org/articles/92404/figures#video2

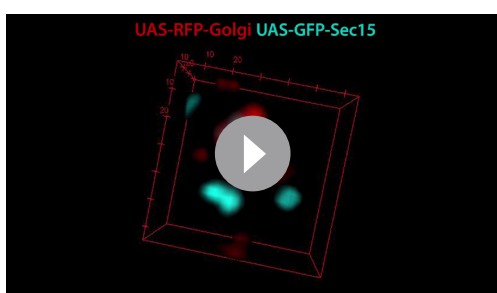

**Video 3.** 3D recontruction of GFP-Sec15 *foci* (cyan) associated to *trans*-Golgi marker (red). https://elifesciences.org/articles/92404/figures#video3

reasoned that other players, perhaps the exocyst, might be involved in Rab1 association or dissociation from the SG membrane. In fact, silencing of Sec5 under conditions that allowed the formation of immature SGs provoked significant reduction of Rab1-YFP on the SG membrane (*Figure 8G–I*), suggesting that the exocyst participated in Rab1 recruitment (*Figure 8M*).

Remarkably, immature SGs that resulted from Sec5 or Sec3 knock-down displayed higher-than-normal levels of YFP-Rab11 around them (*Figure 8J–L* and *Figure 8—figure supplement 2M–O*), suggesting that the exocyst is a negative regulator of Rab11 recruitment. This observation is in apparent contradiction with previous reports by others and us that indicate that Rab11 is required for SG maturation (*de la Riva-Carrasco et al., 2021*; *Ma and Brill, 2021a*; *Neuman et al., 2021*). Noteworthy, overexpression of a constitutively active form of Rab11 (UAS-YFP-Rab11^{CA}), which was readily recruited to SGs, also provoked an arrest of SG maturation, thereby phenocopying exocyst complex knock-down (*Figure 8—figure supplement 3A–C*). Thus, during maturation of SGs, the levels of active Rab11 need to be precisely regulated, and this is achieved, at least in part, by the exocyst (*Figure 8M* and *Figure 8—figure supplement 3F*).

Given that the exocyst is an effector of different Rab-GTPases during vesicle exocytosis (*Guo et al., 1999*; *Novick et al., 1995*; *Wu et al., 2005*), we next investigated if Rab11 is required for recruiting the exocyst to SGs. Indeed, we found that Sec15 failed to localize on SGs following knock-down of Rab11 (*Figure 8—figure supplement 3D, E*), indicating a crucial role of Rab11 in recruitment of Sec15, and probably of the whole exocyst to SGs.

Overall, the results described in this section suggest that Rab11 recruits Sec15, and perhaps the whole exocyst, to immature SGs to allow homotypic fusion and maturation (*Figure 8—figure supplement 3D, E*), while in turn, the exocyts limits the levels and/or activity of Rab11 on SGs (*Figure 8J–L* and *Figure 8—figure supplement 2M–R*), as excessive Rab11 is apparently detrimental for SG maturation (*Figure 8—figure supplement 3A–C*). We propose that a single-negative feedback loop precisely regulates Rab11 and exocyst complex activity/levels, thus controlling recruitment of maturation factors such as Syt-1, CD63, and Rab1, and therefore, the outcome of SG maturation (*Figure 8—figure supplement 3F*).

## The exocyst is required for SG fusion with the apical plasma membrane

Under low silencing conditions of exocyst subunits the prevalent phenotype was mature SGs retained in salivary gland cells (*Figures 3D and 9A*, and *Figure 3—figure supplement 1B*), suggesting a function of the complex in SG fusion with the apical plasma membrane. The process of SG–plasma membrane fusion can be assessed by visualizing the incorporation of plasma membrane-specific components to the membrane of SGs (*de la Riva-Carrasco et al., 2021*; *Rousso et al., 2016*; *Tran et al., 2015*). Phosphatidylinositol 4,5-bisphosphate (PI(4,5)P$_2$) is a lipid of the inner leaflet of the apical plasma membrane, which is absent in endomembranes (*Phan et al., 2019*). Using a fluorophore-based reporter of PI(4,5)P$_2$ the SGs that have fused with the plasma membrane can be distinguished from those that have not. Using this approach, we found that silencing of Exo70 impaired fusion with the plasma membrane (*Figure 9B–D*). Consistent with a role of the exocyst in fusion of SGs with the plasma membrane, at this stage of development GFP-Sec15 was no longer detected between SGs, but rather at sites of contact of SGs with the plasma membrane (*Figure 9E–G* and *Video 7*). These data indicate that during regulated exocytosis, the exocyst complex is required for fusion of SGs with the apical plasma membrane, possibly acting as a tethering complex (*Figure 9H–I*).

Overall, by utilizing the *Drosophila* larval salivary gland, we have made a comprehensive analysis of the role of the exocyst complex in the pathway of regulated exocytosis. We found that the exocyst is critically required for biogenesis of SGs, for their maturation and homotypic fusion and for mediating fusion between SGs and the plasma membrane.

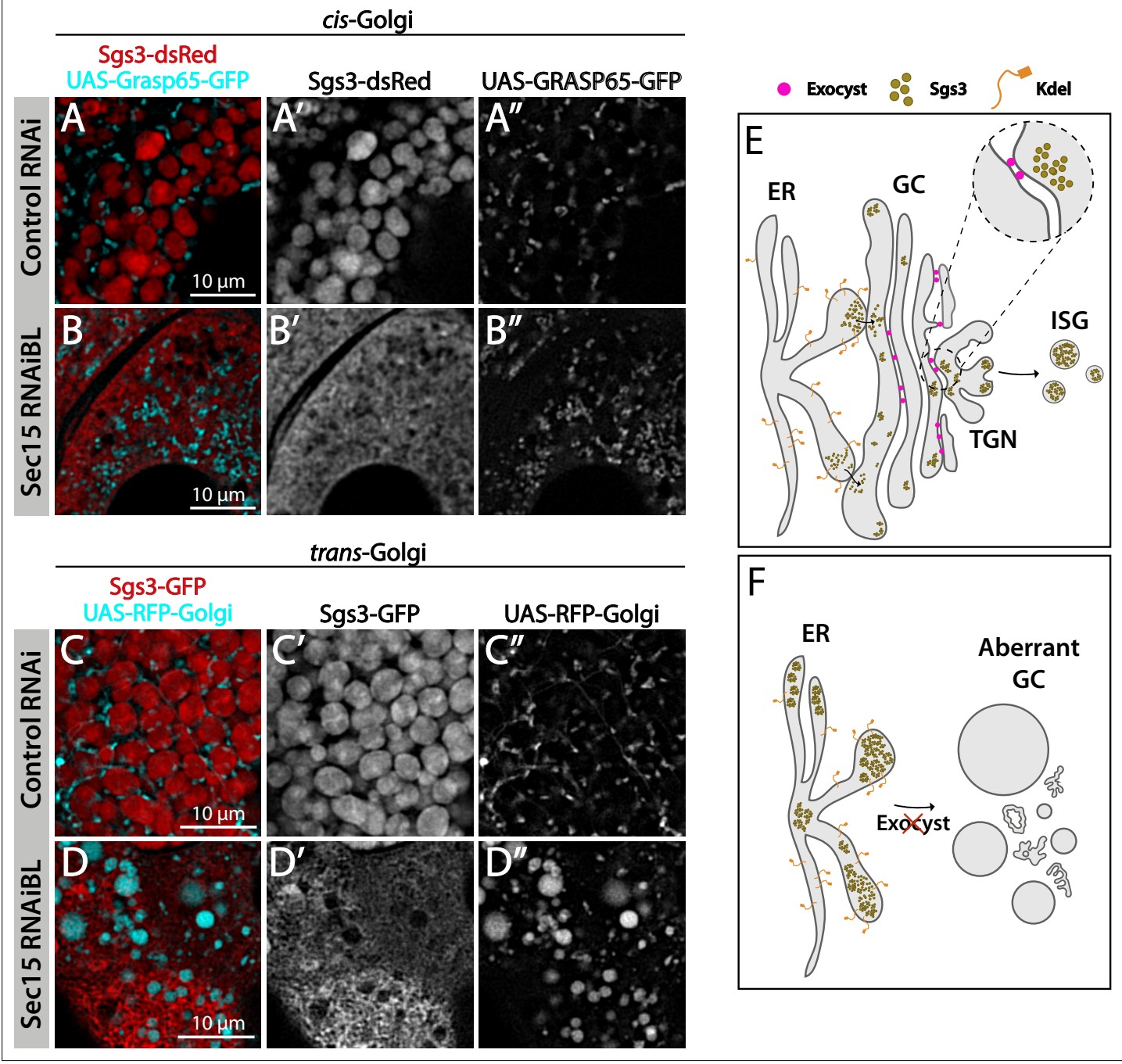

**Figure 6.** The exocyst is required to maintain normal Golgi complex morphology. Confocal images of unfixed salivary glands of the indicated genotypes. Larvae were grown at 29°C to achieve maximal activation of the Gal4-UAS system and maximal downregulation of exocyst subunits. In control larvae (*fkh-Gal4/UAS-white^RNAi*), Sgs3-dsRed (**A**) or Sgs3-GFP (**C**) were in mature secretory granules (SGs). In *sec15^RNAi* salivary glands (*fkh-Gal4/UAS-sec15^RNAiBL*) Sgs3 was retained in a mesh-like structure (**B, D**), also shown in *Figure 4*. (**A, B**) The *cis*-Golgi complex was labeled with Grasp65-GFP (*UAS-Grasp65-GFP*), and the *trans*-Golgi with RFP-Golgi (*UAS-RFP-Golgi*), (**C, D**). The morphology of the *cis*- and *trans*-Golgi complexes changed dramatically in *sec15*-knock-down cells (**B″, D″**), in comparison to controls (**A″, C″**). Transgenes were expressed with *fkh-Gal4*. Scale bar 10 μm. (**E**) Model of Sgs3 transit from the endoplasmic reticulum (ER) through the *cis*- and *trans*-Golgi complex to sprouting SGs from the *trans*-Golgi complex. (**F**) The exocyst is needed for tethering the Golgi complex cisternae and to support Golgi complex structure. In the absence of the exocyst Golgi *cisternae* disconnect, *cis*- and *trans*-Golgi become dysfunctional, resulting in Sgs3 retention at the ER.

The online version of this article includes the following source data and figure supplement(s) for figure 6:

**Figure supplement 1.** The exocyst is required to maintain the typical morphology of the Golgi complex.

**Figure supplement 1—source data 1.** Raw data used to generate *Figure 6—figure supplement 1*.

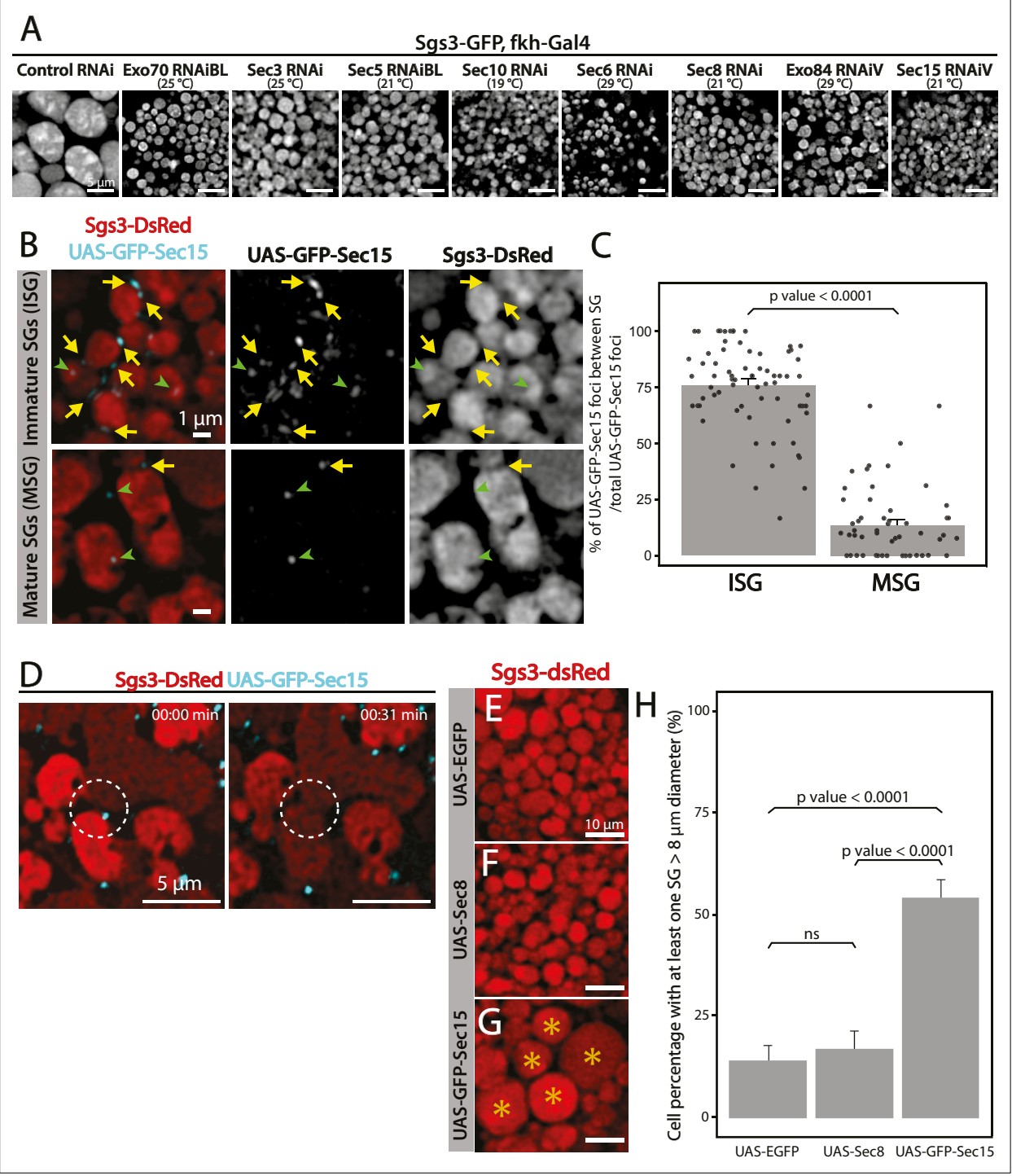

**Figure 7.** The exocyst is required for secretory granule (SG) homotypic fusion. (**A**) Confocal images of unfixed salivary glands of the indicated genotypes. In control salivary glands (*sgs3-GFP, fkh-Gal4/UAS-cherry*[RNAi]), Sgs3-GFP was packed in mature SGs. RNAis targeting any of the subunits of the exocyst were expressed at the indicated temperatures, giving rise to salivary cells with immature granules. Scale bar 5 μm. (**B**) GFP-Sec15 (cyan) and Sgs3-dsRed (red) were expressed in salivary glands (*sgs3-dsRed/UAS-GFP-sec15; fkh-Gal4*) at 18°C. GFP-Sec15 *foci* were mostly found in between SGs (yellow arrows) in cells bearing immature SGs (upper panel) and not in cells with mature SGs (lower panel), in which most *foci* did not localize in between SGs (green arrowheads). Scale bar 1 μm. (**C**) Quantification of Sec15 *foci* in between SGs relative to the total number of Sec15 *foci* (%) (Wald test, p-value <0.05). Eight salivary glands with ISGs and five salivary glands with MSGs were analyzed. (**D**) Still panels of two different frames of *Video 6* showing that during a homotypic fusion event, GFP-Sec15 accumulated precisely at the contact site between two neighboring SGs (dotted circle); scale bar 5 μm. (**E–G**) Expression of EGFP (*sgs3-dsRed; UAS-EGFP, fkh-Gal4*) (**E**) or Sec8 (*sgs3-dsRed/UAS-Sec8; fkh-Gal4*) (**F**) did not affect SG size (red). Expression of

*Figure 7 continued on next page*

*Figure 7 continued*

GFP-Sec15 (**G**) (*sgs3-dsRed/UAS -GFP-sec15; fkh-Gal4*) generated giant SGs (asterisks). Scale bar 10 µm. (**H**) Quantification of the percentage of salivary gland cells with at least one SG larger than 8 µm diameter; (Likelihood ratio test followed by Tukey's test, p-value <0.05). 'n' represents the number of salivary glands analyzed. UAS-EGFP *n* = 7; UAS-Sec8 *n* = 5; UAS-GFP-Sec15 *n* = 5. Transgenes were expressed using *fkh-Gal4.* ns = not significant.

The online version of this article includes the following source data for figure 7:

**Source data 1.** Raw data used to generate *Figure 7*.

## Discussion

The exocyst was initially identified as crucial for secretion in yeast (*Novick et al., 1980*; *TerBush et al., 1996*), and later on, for basolateral trafficking in animal epithelial cells (*Grindstaff et al., 1998*; *Lipschutz et al., 2000*; *Langevin et al., 2005*), with implications in various cellular processes, including cell migration, cytokinesis, ciliogenesis, and autophagy (*Bodemann et al., 2011*; *Park et al., 2010*; *Rogers et al., 2004*; *Thapa et al., 2012*). Recently, an in-depth study of the mammalian exocyst was performed, showing that each of the eight subunits of the complex is essential for constitutive secretion of soluble proteins (*Pereira et al., 2023*). However, fewer is known about the function of the exocyst in regulated exocytosis. A recent work indicates that, in cultured pancreatic beta cells, the exocyst mediates tethering of insulin-containing granules to the plasma membrane and to the cortical F-actin network, being required for exocytosis of a subset of these granules (*Zhao et al., 2023*). Previous studies had implicated specific subunits of the complex in insulin-stimulated exocytosis of the glucose transporter Glut4 in cultured adipocytes (*Inoue et al., 2003*), as well as in skeletal muscle cells (*Fujimoto et al., 2019*). In *Drosophila*, some subunits of the complex (Sec3, Sec5, Sec6, Sec8, and Sec15) were recently suggested to be required for SG development (*Ma et al., 2020*), although earlier, a role for the Sec10 in salivary gland exocytosis had been disregarded (*Andrews et al., 2002*). Therefore, a role of the exocyst in regulated exocytosis is less clear. In this study, not only we demonstrate that the eight subunits of the complex are required for regulated exocytosis of SGs of *Drosophila* larval salivary gland, but also, that the holocomplex participates in multiple steps along the secretory pathway (*Figure 10*).

Most studies of the secretory pathway rely on a bimodal readout: Intracellular retention of SGs versus exocytosis of SG content. These approaches set a limit to our understanding of the specific functions that regulators and effectors exert on the exocytic pathway. By using fluorescently labeled versions of the Sgs3 cargo protein in *Drosophila* salivary glands we have shown that knock-down of any of the exocyst subunits can bring about three distinct phenotypic outcomes: (1) Impairment of SG biogenesis, (2) Impairment of SG homotypic fusion and maturation, or (3) Impairment of SG fusion with the apical plasma membrane. Noteworthy, the frequency at which each of these defects

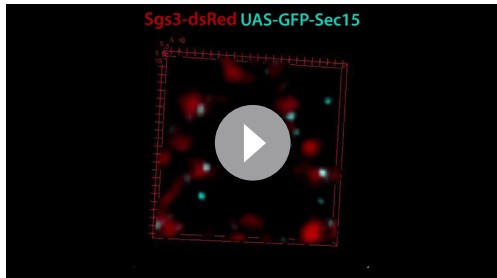

**Video 4.** Three-dimensional reconstruction showing association between the exocyst and immature secretory granules (SGs). A salivary gland from a third instar larva (~186 hr at 18°C equivalent to ~100 hr at 25°C) was imaged. SGs with Sgs3-dsRed (red) and the exocyst marker GFP-Sec15 (cyan) were in close association (*sgs3-dsRed/UAS-GFP-sec15; fkh-Gal4*). Three-dimensional reconstruction of 6 slides with 0.4 µm spacing was generated using ImageJ. Crop size is 10 µm.

https://elifesciences.org/articles/92404/figures#video4

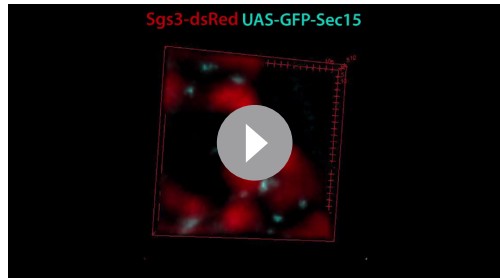

**Video 5.** Three-dimensional reconstruction showing association between the exocyst and immature secretory granules (SGs). A salivary gland from a third instar larva (~186 hr at 18°C equivalent to ~104 hr at 25°C) was imaged. SGs with Sgs3-dsRed (red) and the exocyst marker GFP-Sec15 (cyan) were in close association (*sgs3-dsRed/UAS-GFP-sec15; fkh-Gal4*). Three-dimensional reconstruction of 12 slides with 0.45 µm spacing was generated using ImageJ. Crop size is 10 µm.

https://elifesciences.org/articles/92404/figures#video5

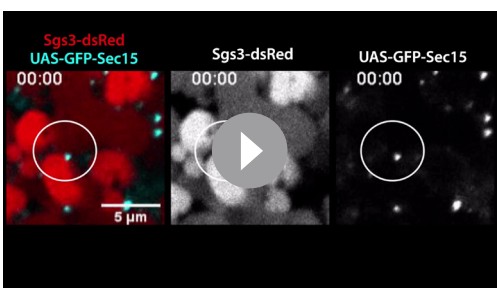

**Video 6.** Real-time imaging of a fusion event between secretory granules (SGs). A *Drosophila* salivary gland expressing Sgs3-dsRed and GFP-Sec15 (*sgs3-dsRed/ UAS-GFP-sec15; fkh-Gal4*) in which a fusion event between SGs was captured (circle). Note that the exocyst (GFP-Sec15) localized at the fusion point between two SGs. The movie was not deconvolved, and represents a maximum intensity projection of three optical slices (total range: 1,6 µm). Scale bar 5 µm. https://elifesciences.org/articles/92404/figures#video6

occurs depends on the extent of exocyst down-regulation, implying that each of the three functions depends on different levels of the complex: SG–plasma membrane fusion is highly sensitive to even slight reduction of exocyst subunits levels; SG maturation requires intermediate levels of the complex, and SG biogenesis seems to be the most robust of the three processes, and strong reduction of exocyst levels is required to provoke this defect.

Although exocyst subcomplexes or individual subunits have been suggested to play specific roles in cell biology (*Mehta et al., 2005*), it is generally believed that the functions of the exocyst are carried out by the holocomplex (*Ahmed et al., 2018*). Studies analyzing the requirement of all eight subunits in a single biological process are not abundant. In *Drosophila*, a study was performed in which the eight subunits were found to be required for developmentally regulated autophagy during salivary cell death, but not for starvation-induced autophagy in the fat body (*Tracy et al., 2016*), and also for proper synaptic development at neuromuscular junctions (*Kang et al., 2024*). However, in other biological settings some subunits of the exocyst (Sec3, Sec5, Sec6, Sec8, and Sec10) but not others (Sec15, Exo70, and Exo84) were shown to be specifically required for general as well as for specific autophagy in yeast (*Singh et al., 2019*). In mammalian cells, Exo84- and Sec5-induced exocyst complex activation have opposing roles on activation of the autophagy machinery (*Bodemann et al., 2011*). Our systematic analysis of the requirement of each of the exocyst subunits in biogenesis, maturation, and exocytosis of SGs led us to conclude that all three biological processes are carried out by the holocomplex, and not by individual subunits or subcomplexes.

Consistent with the three biological functions that the exocyst exerts in the secretory pathway, we found that its subcellular localization during salivary gland development is dynamic (*Figure 10*). Immediately after the onset of Sgs3 synthesis, the exocyst associated with the Golgi complex; later, when SG biogenesis has begun, the exocyst was mostly present at the fusion point between immature SGs, and finally it localized on mature SGs in close proximity to the apical plasma membrane. These dynamic subcellular localization is coherent with the three phenotypes that we have observed after precise modulation of exocyst levels. Exocyst localization at the TGN and at the plasma membrane has been reported before in mammalian cells (*Yeaman et al., 2001*), and although it was suggested in that study that the exocyst might be required at several steps of the secretory pathway from the TGN to the plasma membrane, the idea was not further explored.

By analyzing SG–plasma membrane fusion markers, we have shown that, paralleling the requirement of the exocyst in constitutive exocytosis, the complex participates in SG fusion with the plasma membrane in regulated exocytosis as well. Moreover, we have found that the Sec15 subunit localizes exactly at contact sites between SGs and the plasma membrane, further supporting the notion that the exocyst plays a role in this fusion process. It is accepted that the function exerted by the exocyst in the fusion between SGs and the plasma membrane depends on Ral GTPases (*Brymora et al., 2001*; *Wang et al., 2004*). RalA is the only Ral GTPase described so far in *Drosophila*, and we have recently reported that RalA is involved in SG–plasma membrane fusion in the salivary gland (*de la Riva-Carrasco et al., 2021*). Therefore, it seems likely that, paralleling constitutive exocytosis, the interaction between RalA and the exocyst is critical for tethering SGs to the plasma membrane during regulated exocytosis as well. The molecular components that regulate tethering and fusion of SGs with the plasma membrane remain largely unknown. Perhaps, this is because the proteins involved in these processes participate in earlier steps of the secretory pathway as well. Although other potential roles of the exocyst in salivary gland biology that could tangentially impact SG biogenesis cannot be completely ruled out, our temperature-dependent manipulation of the expression of exocyst subunits,

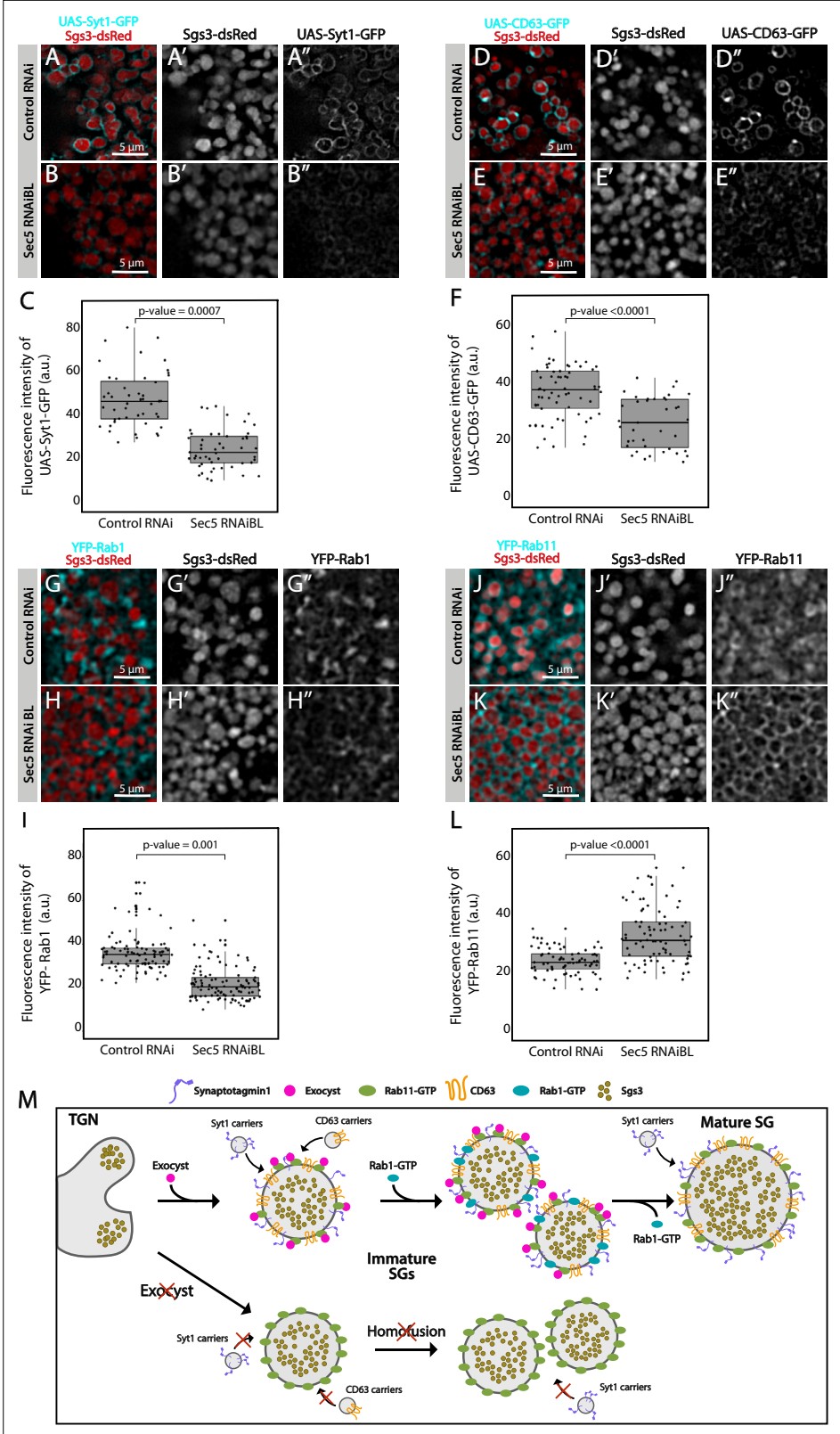

**Figure 8.** The exocyst mediates the acquisition of secretory granule (SG) maturation factors. Confocal images of unfixed salivary glands of the indicated genotypes. Larvae were grown at 21°C to reduce the activity of the Gal4-UAS system, and to generate a maximal proportion of cells with immature SGs. Recruitment of (**A, B**) Syt1-GFP (*UAS-Syt1-GFP*); (**D, E**) CD63-GFP (*UAS-CD63-GFP*); (**G, H**) YFP-Rab1; (**J, K**) YFP-Rab11 was analyzed in control

*Figure 8 continued*

salivary glands expressing *white^RNAi* (*fkh-Gal4/UAS-white^RNAi*) and in salivary glands expressing *sec5^RNAiBL* (*fkh-Gal4/ UAS-sec5^RNAiBL*). Fluorescent intensity around SGs of each of the analyzed maturation factors was quantified using the ImageJ software and plotted (**C, F, I, L**). Comparison of fluorescent intensity among genotypes and statistical analysis was performed using one-way analysis of variance (ANOVA). '*n*' represents the number of salivary glands: (**C**) *control RNAi n = 4, sec5^RNAiBL n = 5*; (**F**) *control RNAi n = 5, sec5^RNAiBL n = 4*; (**I**) *control RNAi n = 6, sec5^RNAiBL n = 9*; (**L**) *control RNAi n = 11, sec5^RNAiBL n = 8*. Transgenes were expressed using *fkh-Gal4*. Scale bar 5 µm. (**M**) Proposed model of exocyst-dependent SG homotypic fusion and maturation. The exocyst complex is required for homotypic fusion between immature SGs. Immature SGs incorporate Syt-1, CD63, and Rab1 in an exocyst-dependent manner. Syt-1 continues to be recruited to SGs after homotypic fusion. The exocyst complex also inhibits the incorporation of an excess of Rab11 around SGs.

The online version of this article includes the following source data and figure supplement(s) for figure 8:

**Source data 1.** Raw data used to generate *Figure 8*.

**Figure supplement 1.** Incorporation of maturation markers to secretory granules (SGs) in wild-type salivary glands.

**Figure supplement 2.** The exocyst mediates the acquisition of secretory granule (SG) maturation factors.

**Figure supplement 2—source data 1.** Raw data used to generate *Figure 8—figure supplement 2O and R*.

**Figure supplement 3.** The exocyst negatively regulates incorporation of Rab11 to secretory granules (SGs).

**Figure supplement 3—source data 1.** Raw data used to generate *Figure 8—figure supplement 3C*.

in combination with differential requirements of the complex in terms of quantity/activity in each of the processes was instrumental for uncovering three different discrete steps in which the exocyst plays a role along the regulated secretory pathway.

Proteins that will be secreted are co-translationally translocated to the ER, and then transported into the *cis*-region of the GC in COPII-coated vesicles that are 60–90 nm in diameter and therefore, sufficient to accommodate most membrane and secreted molecules (*Raote and Malhotra, 2021*). Larger cargos, such as collagen and mucins might use alternative ER-*cis*-GC communication mechanism that are independent of vesicular carriers. Specifically, direct connections between the ER and *cis*-GC are formed in a Tango1- and COPII-dependent manner (*Reynolds et al., 2019*; *Yang et al., 2024*). Disruption of these connections not only affects SG formation, but also has profound impact on GC structure (*Bard and Malhotra, 2006*; *Ríos-Barrera et al., 2017*). We found that strong silencing of any of the exocyst subunits results in retention of Sgs3 at the ER, abrogation of SG biogenesis, and alteration of normal morphology of the GC. In *Drosophila*, as well as in mammalian cells, the Golgi complex is polarized in cis and *trans*-cisternae that are held together by tethering complexes of the CATCHR family. Disruption of these connections results in altered Golgi complex morphology, reflected in fragmentation and swelling of *cisternae*, and impairment of the secretory pathway (*D'Souza et al., 2020*; *Khakurel and Lupashin, 2023*; *Liu et al., 2019*). The fact that exocyst silencing not only affected SG formation, but also had profound impact on GC structure, together with the observation that Sec15 specifically associated with the Golgi complex but not with the ER before the onset of SG biogenesis, argues in favor of a role of the exocyst as a tethering complex between cisternae. We propose that the exocyst might be redundant with other CATCHR complexes in this function, since only severe downregulation of exocyst subunits expression can manifest this phenotype.

Maturation of SGs is a multidimensional process that involves homotypic fusion, acidification, cargo condensation, and acquisition of membrane proteins that will steer SGs to the apical plasma membrane, and contribute to recognition, tethering, and fusion (*Boda et al., 2023*; *Ji et al., 2018*; *Nagy et al., 2022*; *Syed et al., 2022*). Previously, the exocyst was shown to be required for maturation of Weibel–Palade bodies (*Sharda et al., 2020*). In the current work, we have shown that the exocyst participates in several aspects of SG maturation. One of such processes is homotypic fusion of immature SGs. We found that an adequate extent of downregulation of the expression of any of the exocyst subunits leads to accumulation of immature SGs. Furthermore, consistent with a function of the exocyst in SG homofusion, we found that Sec15 localizes preferentially at the fusion point between adjacent immature SGs. In support to this notion, we found that overexpression of Sec15 results in unusually large granules, likely derived from uncontrolled homotypic fusion. Therefore, our data suggest that the exocyst complex is the tethering factor responsible to bring immature SGs in close proximity to enable homofusion.

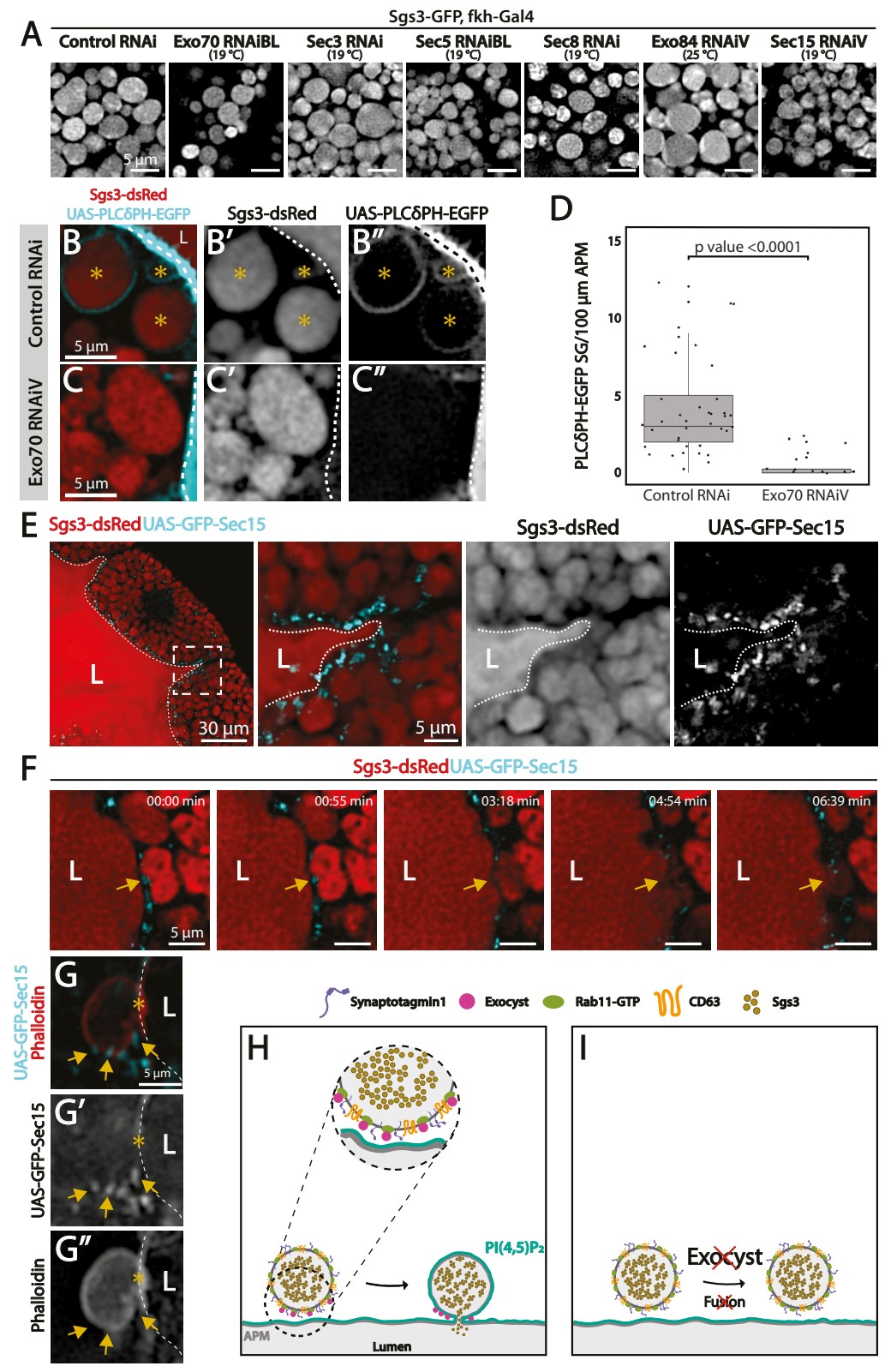

**Figure 9.** The exocyst is required for secretory granule (SG) fusion with the plasma membrane during regulated exocytosis. Confocal images of unfixed salivary glands of the indicated genotypes. Larvae were grown at the indicated temperatures to attain levels of RNAi-mediated silencing that bring about maximal proportion of cells with mature, exocytosis incompetent SGs. (**A**) Sgs3-GFP localized within SGs. Control SGs (*sgs3-GFP, fkh-Gal4/*

*Figure 9 continued*

*UAS-cherry*^*RNAi*) were indistinguishable from SGs of salivary cells in which a subunit of the exocyst has been knocked-down. Scale bar 5 µm. (**B**) In control salivary glands (*fkh-Gal4/UAS-white*^*RNAi*), the PI(4,5)P$_2$ reporter *UAS-PLCγ-PH-GFP* labels the plasma membrane (dotted line) and also the SGs that have already fused with the plasma membrane (asterisks). (**C**) SGs of cells expressing *exo70*^*RNAiV*^ (*UAS-exo70*^*RNAiV*^; *fkh-Gal4*) were not labeled with the reporter, indicating that these SGs failed to fuse with the plasma membrane. Scale bar 5 µm. (**D**) The number of mature SGs positive for PLCγ-PH-EGFP per 100 µm of linear plasma membrane was quantified in the indicated genotypes; Exo70 knock-down reduced SG–plasma membrane fusion Wald test (p-value <0.05; 7 salivary glands per genotype were analyzed. (**E**) During SG exocytosis the exocyst complex, labeled with GFP-Sec15 (cyan) localized as dots in contact sites between SGs and the apical plasma membrane (dotted line) (*sgs3-dsRed/UAS-GFP-sec15; fkh-Gal4*). (**F**) Still panels of *Video 7* showing a fusion event between a mature SG and the plasma membrane (arrow); a dot of GFP-Sec15 indicating the position of the exocyst (arrow) was positioned just at the site where fusion was taking place. Scale bar 5 µm. (**G**) Confocal image of a fixed salivary gland. A mature SG that has fused with the plasma membrane (dotted line), and thus became labeled with phalloidin, displayed dots of GFP-Sec15 on its side (arrow), just next to the fusion point (asterisk). Transgenes were expressed with *fkh-Gal4*. The salivary gland lumen is indicated with 'L'. Scale bar 5 µm. (**H**) Model of the role of the exocyst complex during SG–plasma membrane fusion during regulated exocytosis. The exocyst sits on the membrane of the SG, and tethers the granule to the plasma membrane, favoring the action of fusion molecules. (**I**) Upon loss of the exocyst complex, mature SGs cannot contact the plasma membrane and fusion does not occur.

The online version of this article includes the following source data for figure 9:

**Source data 1.** Raw data used to generate *Figure 9*.

---

Besides SG growth by homotypic fusion, several factors are recruited to maturing SGs, including the transmembrane proteins CD63 and Syt-1. Our temperature-dependent genetic manipulations revealed that the addition of Syt-1 to SGs occurs immediately after they have emerged from the TGN, and addition of this protein continues after they have attained a mature size. Syt-1 localizes at the basolateral plasma membrane of salivary gland cells before SG biogenesis, and it is not known how Syt-1 reaches maturing SGs. Given that the exocyst is required for loading SGs with Syt-1, one possibility is that a vesicular carrier transports Syt-1 from the basolateral membrane to nascent and maturing SGs, and that the contact between SGs and these vesicles is mediated by the exocyst. CD63, which is also required for SG maturation, localizes instead at the apical plasma membrane before SG biogenesis, and reaches the SGs from the endosomal retrograde pathway through endosomal tubes (*Ma et al., 2020*). We found that cells in which the exocyst has been knocked-down display reduced levels of CD63 around SGs, suggesting that the exocyst might contribute to the contact between endosomal tubes and SGs as well. Based on our analysis, this maturation event should take place mostly before, and not after, homotypic fusion. Thus, our results support the notion that the exocyst complex might contribute to tethering maturing SGs to different types of membrane-bound carriers that transport maturation factors like Syt-1 and CD63. These observations expand the notion that a crosstalk between the secretory and endosomal pathways occurs (*Ma and Brill, 2021a*; *Ma et al., 2020*; *Papandreou and Tavernarakis, 2020*), and that the exocyst is a critical factor linking the two pathways.

Vesicle maturation, either at the endocytic or the exocytic pathways, usually involves changes in vesicle-bound Rab proteins (*Ailion et al., 2014*; *Ma and Brill, 2021a*; *Thomas et al., 2021*). The mechanisms that mediate these Rab-switches are not completely understood. Specifically, in *Drosophila* larval salivary glands, Rab1 and Rab11 association to SGs appear to be crucial for SG maturation and timely exocytosis (*Neuman et al., 2021*). We found that the levels of Rab1 and Rab11 on SGs are, respectively, positively and

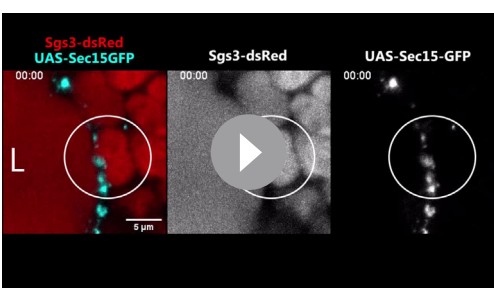

**Video 7.** Real-time imaging of a fusion event between a secretory granule (SG) and the plasma membrane. A *Drosophila* salivary gland expressing Sgs3-dsRed and GFP-Sec15 (*sgs3-dsRed/UAS-GFP-sec15; fkh-Gal4*) in which a fusion event between SGs and the plasma membrane was captured (circle). The granule content was released to the gland lumen (L). The movie shows a single slice of 32 frames comprising a total time of 0.32 s. The movie was not deconvolved. Scale bar 5 µm.
https://elifesciences.org/articles/92404/figures#video7

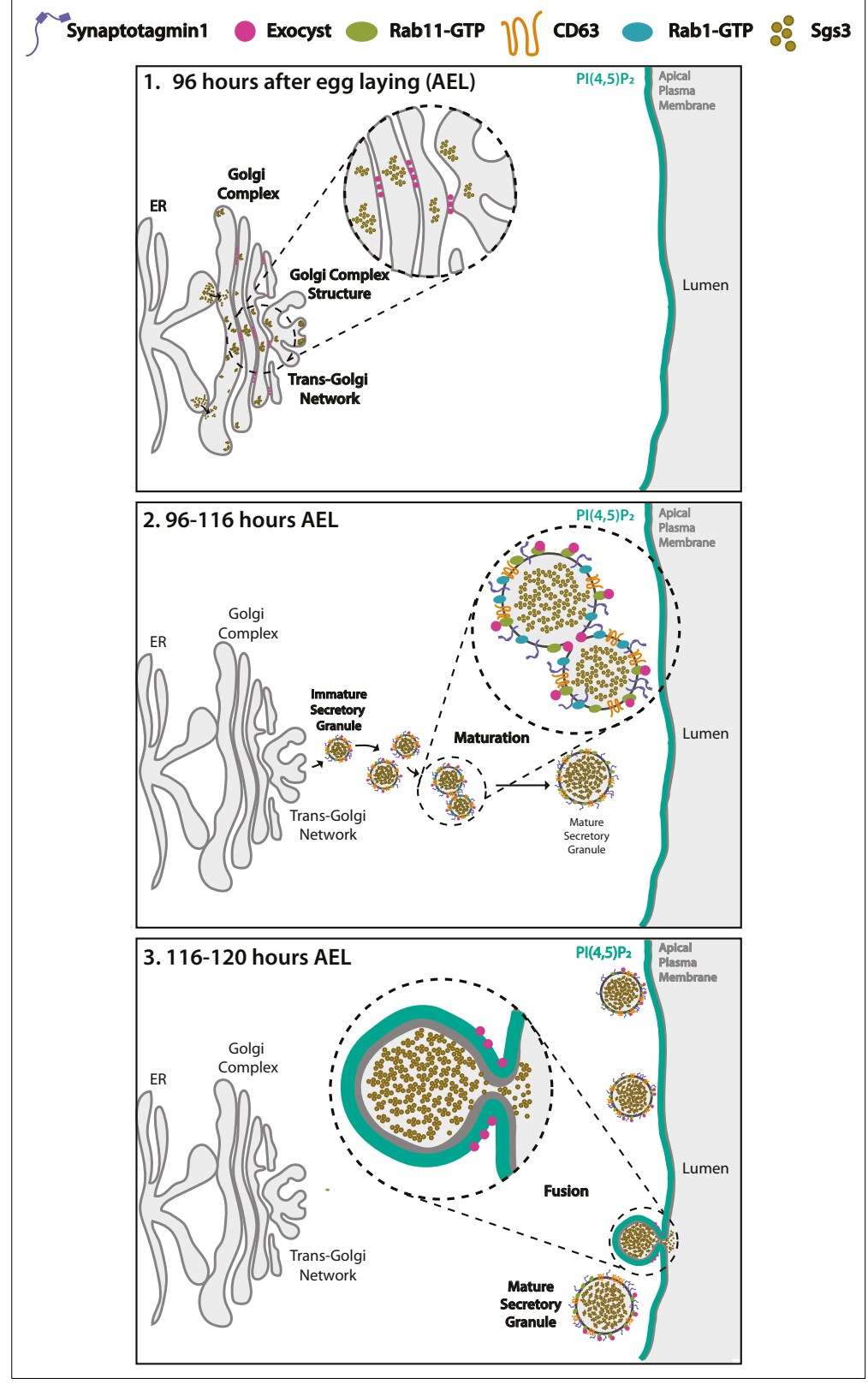

**Figure 10.** Proposed model of the action of the exocyst in maintenance of normal Golgi complex structure, maturation, and exocytosis of secretory granules (SGs) in *Drosophila* larval salivary gland cells. (1) Before SG biogenesis (<96 h AEL), the exocyst (pink dots) localizes at the Golgi complex, where it is required to maintain the normal Golgi structure. The mucine Sgs3 (brown dots) moves through the secretory pathway from the endoplasmic

*Figure 10 continued on next page*

*Figure 10 continued*

reticulum to the Golgi complex, from where immature SGs containing the mucine sprout out. (2) After sprouting, SGs undergo maturation (96–116 h AEL). During maturation, the exocyst localizes in between immature SGs, where it is required for homotypic fusion. The exocyst is also required for incorporation of maturation factors to the membrane of SGs. These maturation factors include Syt-1 (purple line), DC63 (orange line), Rab11 (green oval), and Rab1 (light blue oval). At this stage, the exocyst no longer localizes at the Golgi complex. (3) When maturation has been completed SGs fuse with the apical plasma membrane and exocytosis takes place. During exocytosis (116–120 h AEL), the exocyst localizes at mature SGs, in contact with the apical plasma membrane (APM), where it is required for tethering and subsequent fusion, prior to release of the SG content to the salivary gland lumen.

negatively regulated by the exocyst, and that additionally, Rab11 is itself required for recruitment of Sec15 to immature SGs. These intrincated relationships between Rab11, Rab1 and the exocyst can be explained through a single-negative feedback loop that would guarantee adequate levels of Rab11 and the exocyst on maturing SGs. This possibility is supported by previous evidence from *Drosophila* and other systems that indicate that Rab11 and Sec15 interact physically and genetically (*Escrevente et al., 2021*; *Guo et al., 1999*; *Novick, 2016*). In fact, the exocyst is an effector of Rab11 (Sec4p in yeasts), as GTP-bound Rab11 recruits Sec15 to secretory vesicles, which in turn triggers formation of the holocomplex on the vesicle membrane as previously shown in yeast, *Drosophila* and mammalian cells (*Wu et al., 2005*; *Takahashi et al., 2012*; *Zhang et al., 2004*).

An increasing number of publications reveal the complexity and variety of vesicle trafficking routes that feed into the biogenesis, maturation and exocytosis of SGs. Our study has revealed that the exocyst holocomplex acts at several steps of the pathway of regulated exocytosis, maintaining Golgi complex morphology to allow protein exit from the ER, and transit through the early secretory pathway; promoting homotypic fusion of SGs and acquisition of maturation factors; and finally, allowing SG–plasma membrane fusion (*Figure 10*). We propose that all these events depend on the activity of the exocyst complex as a tethering factor.

## Materials and methods
### Fly stocks and genetics
All fly stocks were kept on standard corn meal/agar medium at 25°C. Crosses were set up in vials containing two males and five females of the required genotypes at 25°C. Crosses were flipped every 24 hr to avoid larval overcrowding, and then moved to the desired temperature: 29, 25, 21, 19, or 18°C. The temperature used for each experiment is specified in figure legends. In the experiments of *Figures 2, 3 and 8* and *Figure 3—figure supplement 1* and *Figure 8—figure supplement 2*, all crosses were maintained in a bath water to reduce temperature fluctuations. *D. melanogaster* lines used in this work are listed in *Table 2*, and were obtained from the Bloomington *Drosophila* Stock Center (http://flystocks.bio.indiana.edu) or from the Vienna *Drosophila* Stock Center (https://stock-center.vdrc.at); Sgs3-dsRed was generated by A.J. Andres' Lab (University of Nevada, United States).

### Sgs3-GFP retention phenotype
Larvae or prepupae of the desired genotype were visualized and photographed inside glass vials under a fluorescence dissection microscope Olympus MVX10. In prepupae, localization of Sgs3-GFP or Sgs3-dsRed inside salivary glands or outside the puparium was determined. Each experiment was repeated at least three times.

### Developmental staging and SG size
When larvae were cultured at 25°C, SG maturation progressed according to the timeline shown in *Figure 1*. Roughly, at 29°C larval development was shortened by 24 hr, and extended in 1, 3, and 4 days when larvae were cultured at 21, 19, and 18°C, respectively. Precise physiological staging of each salivary gland was carried out according to *Neuman et al., 2021* upon dissection and observation under the confocal microscope. In all experiments, only SGs from the distal-most cells of the salivary glands were imaged to avoid potential variations in SG size due to desynchronization in the synthesis of Sgs3. Data of temperatures and developmental staging from each experiment are synthesized in *Table 4*.

**Table 4.** Salivary gland developmental staging.

Salivary glands analyzed in experiments of the indicated figures. Columns display: Experimental temperature, larval hours of development after egg laying (AEL), equivalent hours of development at 25°C (based on SG phenotype and salivary gland general appearance), presence or absence of Sgs3 in salivary glands, at the developmental time studied, and expected stage of SGs (mature or immature).

| | Experimental temperature (°C) | Hours of development at the experimental temperature (h AEL) | Equivalent hours of development at 25°C (h AEL) | Presence of Sgs3 | Expected developmental stage of SGs in wild type |
|---|---|---|---|---|---|
| *Figure 2* | 29 | ~96 | 116–120 | Yes | Mature |
| *Figure 3* | 19, 21, 25, 29 | ~216, ~144, ~120, ~96 | 116–120 | Yes | Mature |
| *Figure 4* | 29 | ~96 | 116–120 | Yes | Mature |
| *Figure 5* | 18 | ~120–168 | 72–96 | No | - |
| *Figure 6* | 29 | ~96 | 116–120 | Yes | Mature |
| *Figure 7A, E–H* | 29 | ~96 | 116–120 | Yes | Mature |
| *Figure 8A, D, G, J* | 21 | ~108 | 96–104 | Yes | Immature |
| *Figure 8B, E, H, K* | 21 | ~144 | 116–120 | Yes | Mature |
| *Figure 9A–D* | 29 | ~96 | 116–120 | Yes | Mature |
| *Figure 9E–G* | 18 | ~240 | 116–120 | Yes | Mature |
| *Figure 3— figure supplement 1* | 29 | ~96 | 116–120 | Yes | Mature |
| *Figure 3— figure supplement 2* | 19, 25, or 29 | ~216, ~120, ~96 | 112–120 | Yes | Mature |
| *Figure 3— figure supplement 3A, D, G* | 29 | ~96 | 116–120 | Yes | Mature |
| *Figure 3— figure supplement 3B, E, H* | 120 hr at 18°C and 36 hr at 29°C | ~156 | 116–120 | Yes | Mature |
| *Figure 3— figure supplement 4* | 18 | ~240 | 116–120 | Yes | Mature |
| *Figure 3— figure supplement 5* | 25 | ~72 | ~72 | No | - |
| *Figure 3— figure supplement 6* | 25 | ~72 | ~72 | No | - |
| *Figure 3— figure supplement 7A, E* | 25 or 29 | ~72 or ~50 | ~72 | No | - |

*Table 4 continued on next page*

*Table 4 continued*

| | Experimental temperature (°C) | Hours of development at the experimental temperature (h AEL) | Equivalent hours of development at 25°C (h AEL) | Presence of Sgs3 | Expected developmental stage of SGs in wild type |
|---|---|---|---|---|---|
| *Figure 3—figure supplement 7B–D* | 25 | ~72 | ~72 | No | - |
| *Figure 4—figure supplement 1* | 29 | ~96 | 116–120 | Yes | Mature |
| *Figure 4—figure supplement 2* | 29 | ~96 | 116–120 | Yes | Mature |
| *Figure 6—figure supplement 1* | 29 | ~96 | 116–120 | Yes | Mature |
| *Figure 8—figure supplement 1A, E* | 25 | 72–96 | 72–96 | No | - |
| *Figure 8—figure supplement 1B, F* | 25 | 100–104 | 100–104 | Yes | Immature |
| *Figure 8—figure supplement 1C, G* | 25 | 108–112 | 108–112 | Yes | Mature |
| *Figure 8—figure supplement 1D, H* | 25 | 116–120 | 116–120 | Yes | Mature |
| *Figure 8—figure supplement 2A, G, M* | 25 or 21 | 96–104 or ~108 | 96–104 | Yes | Immature |
| *Figure 8—figure supplement 2B, H, N* | 25 or 21 | 116–120 or ~144 | 116–120 | Yes | Mature |
| *Figure 8—figure supplement 2D, E, J, K, P, Q* | 25 or 21 | 116–120 or ~144 | 116–120 | Yes | Mature |
| *Figure 8—figure supplement 3A, B, E* | 29 | ~96 | 116–120 | Yes | Mature |
| *Figure 8—figure supplement 3D* | 29 | ~72 | 96–104 | Yes | Immature |

*Figure 3—figure supplement 1A*

| Genotype | Temperature | Number of glands analyzed | Number of distal cells analyzed | Phenotype | % of phenotype | Standard deviation |
|---|---|---|---|---|---|---|

*Table 4 continued on next page*

*Table 4 continued*

| | Experimental temperature (°C) | Hours of development at the experimental temperature (h AEL) | Equivalent hours of development at 25°C (h AEL) | Presence of Sgs3 | Expected developmental stage of SGs in wild type | |
|---|---|---|---|---|---|---|
| | | | | Mesh-like structure | 0 | 0 |
| | | | | SG immature | 0 | 0 |
| | 29 | 4 | 12 | SG mature | 100 | 0 |
| | | | | Mesh-like structure | 0 | 0 |
| | | | | SG immature | 0 | 0 |
| | 25 | 5 | 13 | SG mature | 100 | 0 |
| | | | | Mesh-like structure | 0 | 0 |
| | | | | SG immature | 0 | 0 |
| | 21 | 5 | 17 | SG mature | 100 | 0 |
| | | | | Mesh-like structure | 0 | 0 |
| | | | | SG immature | 0 | 0 |
| Control RNAi | 19 | 5 | 15 | SG mature | 100 | 0 |
| | | | | Mesh-like structure | 0 | 0 |
| | | | | SG immature | 39.43 | 42.11 |
| | 29 | 5 | 17 | SG mature | 60.57 | 42.11 |
| | | | | Mesh-like structure | 0 | 0 |
| | | | | SG immature | 8.33 | 20.41 |
| | 25 | 6 | 15 | SG mature | 91.67 | 20.41 |
| | | | | Mesh-like structure | 0 | 0 |
| | | | | SG immature | 0 | 0 |
| | 21 | 6 | 17 | SG mature | 100 | 0 |
| | | | | Mesh-like structure | 0 | 0 |
| | | | | SG immature | 0 | 0 |
| Exo70 RNAi V | 19 | 4 | 15 | SG mature | 100 | 0 |

*Table 4 continued*

| | Experimental temperature (°C) | Hours of development at the experimental temperature (h AEL) | Equivalent hours of development at 25°C (h AEL) | Presence of Sgs3 | Expected developmental stage of SGs in wild type | |
|---|---|---|---|---|---|---|
| | | | | Mesh-like structure | 53.33 | 50.55 |
| | | | | SG immature | 46.67 | 50.55 |
| | 29 | 5 | 25 | SG mature | 0 | 0 |
| | | | | Mesh-like structure | 20 | 27.39 |
| | | | | SG immature | 70 | 27.39 |
| | 25 | 5 | 12 | SG mature | 10 | 22.36 |
| | | | | Mesh-like structure | 8.33 | 20.41 |
| | | | | SG immature | 87.5 | 20.92 |
| | 21 | 6 | 21 | SG mature | 4.17 | 10.21 |
| | | | | Mesh-like structure | 5.56 | 13.61 |
| | | | | SG immature | 91.11 | 14.4 |
| Sec5 RNAi V | 19 | 6 | 27 | SG mature | 3.33 | 8.16 |
| | | | | Mesh-like structure | 93.2 | 12.85 |
| | | | | SG immature | 6.8 | 12.85 |
| | 29 | 7 | 38 | SG mature | 0 | 0 |
| | | | | Mesh-like structure | 90 | 22.36 |
| | | | | SG immature | 10 | 22.36 |
| | 25 | 5 | 22 | SG mature | 0 | 0 |
| | | | | Mesh-like structure | 95.83 | 10.21 |
| | | | | SG immature | 4.17 | 10.21 |
| | 21 | 6 | 27 | SG mature | 0 | 0 |
| | | | | Mesh-like structure | 66.43 | 41.31 |
| | | | | SG immature | 33.57 | 41.31 |
| Exo84 RNAi BL | 19 | 5 | 21 | SG mature | 0 | 0 |

*Table 4 continued on next page*

*Table 4 continued*

| Experimental temperature (°C) | Hours of development at the experimental temperature (h AEL) | Equivalent hours of development at 25°C (h AEL) | Presence of Sgs3 | Expected developmental stage of SGs in wild type | |
|---|---|---|---|---|---|
| | | | Mesh-like structure | 100 | 0 |
| | | | SG immature | 0 | 0 |
| 29 | 5 | 25 | SG mature | 0 | 0 |
| | | | Mesh-like structure | 94.44 | 13.61 |
| | | | SG immature | 5.56 | 13.61 |
| 25 | 6 | 26 | SG mature | 0 | 0 |
| | | | Mesh-like structure | 100 | 0 |
| | | | SG immature | 0 | 0 |
| 21 | 5 | 14 | SG mature | 0 | 0 |
| | | | Mesh-like structure | 100 | 0 |
| | | | SG immature | 0 | 0 |
| Sec15 RNAi BL | 19 | 5 | 20 | SG mature | 0 | 0 |

## RNA extraction and cDNA synthesis

Total RNA was isolated from dissected salivary glands of third instar larvae using 500 μl of Quick-Zol reagent (Kalium Technologies, RA00201) following the manufacturer's instructions. The concentration and integrity of the RNA were determined using NanoDrop (Thermo Fisher Scientific) spectrophotometry. RNA (1 μg) was reverse-transcribed using M-MLV Reverse Transcriptase (Invitrogen, 10338842) using oligo-dT as a primer (https://doi.org/10.1080/15548627.2021.1991191). Control reactions omitting reverse transcriptase were used to assess the absence of contaminating genomic DNA in the RNA samples. An additional control without RNA was included.

## Real-time PCR

Gene expression was analyzed by quantitative PCR in a CFX96 Touch (Bio-Rad) cycler. The reactions were performed using HOT FIREPol EvaGreen qPCR Mix Plus (without ROX; Solis BioDyne, 08-25-00001), 0.40 μM primers, and 12–25 ng of cDNA, in a final volume of 10.4 μl (https://doi.org/10.1080/15548627.2021.1991191). Cycle conditions were initial denaturation at 95°C for 15 min, and 40 cycles of denaturation at 95°C for 20 s, annealing at 60°C for 1 min, and extension and optical reading stage at 72°C for 30 s, followed by a dissociation curve consisting of ramping the temperature from 65 to 95°C while continuously collecting fluorescence data (https://doi.org/10.1080/15548627.2021.1991191). Product purity was confirmed by agarose gel electrophoresis. Relative gene expression levels were calculated according to the comparative cycle threshold (CT) method (https://doi.org/10.1080/15548627.2021.1991191). Normalized target gene expression relative to rpl29 was obtained by calculating the difference in CT values, the relative change in target transcripts being computed as $2^{-\Delta CT}$. The efficiencies of each target and housekeeping gene amplification were measured and shown to be approximately equal. Oligonucleotides were obtained from Macrogen (Seoul, Korea), and their sequences were the following: exo70: Fw 5'-GAAGTGGTTCTCCGATCGCT-3', Rv 5'-ACGAGCGG AGGTTGTCTTTT-3'; sec3: Fw 5'-GAAGACGCAACACATGGACG-3', Rv 5'-CTTTGCATATTGGCCC CATCC-3'; sec5: Fw 5'-GTCAATGAGACTGCCAAGAACT-3', Rv 5'-CCTGCAGTGGAATGTGCCTA-3';

rpl29: Fw 5'-GAACAAGAAGGCCCATCGTA-3', Rv 5'-AGTAAACAGGCTTTGGCTTGC-3'. Rpl29 was used as housekeeping gene. Specificity and quality of oligonucleotide sequences for exo70, sec3, sec5, and rpl29 were checked using Primer Blast Resource of the NCBI (http://www.ncbi.nlm.nih.gov/tools/primer-blast/).

## Quantification of the penetrance of phenotypes upon knock-down of exocyst subunits

Sgs3-GFP intracellular distribution was analyzed in salivary gland cells, and one of three phenotypic categories was defined for each cell: (1) 'mesh-like structure' when Sgs3-GFP was distributed in a network-like compartment; (2) 'Immature SGs' when Sgs3-GFP was in SGs with a median diameter smaller than 3 μm; and (3) 'Mature SGs' when Sgs3-GFP was in SGs with a median diameter equal or larger than 3 μm. The penetrance of each of the three phenotypes was calculated for each genotype of interest at the four different temperatures analyzed.

## Salivary gland imaging

Salivary glands of the desired stage were dissected in cold phosphate-buffered saline (PBS) (137 mM NaCl, 2.7 mM KCl, 4.3 mM $Na_2HPO_4$, 1.47 mM $KH_2PO_4$, [pH 8]), and then imaged directly under the confocal microscope without fixation for no more than 5 min. In the experiment of *Figure 9G* and *Figure 3—figure supplement 6E*, salivary glands were fixed for 2 hr in 4% paraformaldehyde (Sigma) at room temperature, and then washed three times for 15 min with PBS-0.1% Triton X-100. For filamentous actin staining, salivary glands were incubated for 1 hr with Alexa Fluor 647 Phalloidin (Thermo Fisher Scientific 1:400) in PBS-0.1% Triton X-100; stained tissues were mounted in gelvatol mounting medium (Sigma) and imaged at a confocal microscope Carl Zeiss LSM 710 with a Plan-Apochromat 63×/1.4NA oil objective, or Carl Zeiss LSM 880 with a Plan-Apochromat 20×/0.8 NA air objective or a Plan-Apochromat 63×/1.4NA oil objective.

For live imaging, salivary glands were dissected in PBS, and then mounted in a 15-mm diameter plastic chamber with a glass bottom made of a cover slip, containing 40 μl of HL3.1 medium (70 mM NaCl, 5 mM KCl, 1.5 mM $CaCl_2$, 2 mM $MgCl_2$, 5 mM HEPES (4-(2-hydroxyethyl)-1-piperazineethanesulfonic acid), 115 mM sucrose, 5 mM trehalose, and pH 7.2 with $NaHCO_3$). The medium was removed allowing the tissue to adhere to the bottom of the chamber, and then a Biopore membrane hydrophilic PTFE (polytetrafluoroethylene) with a pore size of 0.4 μm (Millipore, Sigma) was placed over the sample, and HL3.1 medium was added on top covering the membrane. Images were captured under an inverted Carl Zeiss LSM 880 confocal microscope with a Plan-Apochromat 20×/0.8 NA air objective or a Plan-Apochromat 63×/1.4NA oil objective. For *Video 1*, frames were obtained every 2.25 s, while for *Videos 6 and 7*, frames were captured every 0.78 and 0.67 s, respectively.

## Image processing and analysis

Image deconvolution was performed using the Parallel Spectral Deconvolution plugin of the ImageJ software (NIH, Bethesda, MD) with standard pre-sets (*Schneider et al., 2012*). Image analyses were made with ImageJ (*Schneider et al., 2012*), and graphs were generated with the R Studio software (*R Development Core Team, 2020*). For SG quantification, a region of interest (ROI) from each cell was used. In each ROI, the area of SGs was assessed, and SG diameter was calculated assuming that SGs are circular, using the formula $((Area/\pi)^{1/2})*2$ = diameter. In experiments of *Figure 4B, D, F, H* and *Figure 4—figure supplement 1A, C, E, G, I*, two-dimensional lines scans were generated with the ImageJ plot profile. Fluorescence intensity was determined related to maximal intensity of each marker, always within a linear range. For quantification of fluorescence intensity of Syt1, CD63, Rab1, and Rab11 (*Figure 8* and *Figure 8—figure supplement 2*), the mean intensity of three different ROIs of 5 $μm^2$ from each cell was measured. 17–35 cells from 4 to 11 salivary glands were used in the analysis. Association analyses in *Figure 5B, E* were performed considering structures as associated when the fluorescence spikes, of each fluorophore, had a distance minor or equal to ~0.6 μm. In the experiment of *Figure 7C*, GFP-Sec15 *foci* between granules were measured over total foci in four different ROIs of 225 $μm^2$ from each cell. For nucleus quantification (*Figure 3—figure supplement 6A*) the area from each one, was measured using ImageJ freehand section.

In polarity experiments, the mean fluorescence intensity of apical markers (*Figure 3—figure supplement 7A, B*) was obtained by drawing a plotted line, of ~2 μm wide, over the apical membrane. In the

case of $PI(4,5)P_2$ and F-actin (*Figure 3—figure supplement 7D, E*), localized in both compartments particularly enriched in apical membrane, a perpendicular line, of ~9 μm wide, was draw from the basolateral region to the apical and the coefficient between the maximum fluorescence intensity of both regions was calculated.

Pearson's coefficient was obtained with JACoP plugging of ImageJ (*Bolte and Cordelières, 2006*). The Golgi defect, upon exocyst knock down (*Figure 4—figure supplement 2a*), was measured as the number of cells with aberrant *cis*-Golgi or with swollen *trans*-Golgi vesicles over total cells within each salivary gland.

## Statistical analyses

Statistical significance was calculated using one-way analysis of variance, a Likelihood ratio test or a Wald test, and followed by a Tukey's test with a 95% confidence interval ($p < 0.05$) when comparing multiple factors.

## Acknowledgements

We are grateful to Dr. Andrew Andres, Dr. Gabor Juhasz, the Bloomington Stock Centre, and the Vienna *Drosophila* Resource Centre for fly strains. Dr. Andrés Rossi and Dr Esteban Miglietta, from FIL microscopy facility, for technical support with confocal microscopy; Andrés Liceri for fly food preparation; the FIL personnel for assistance and members of the Wappner lab for fruitful discussions.This work was supported by grants from Agencia Nacional de Promoción de Científica y Tecnológica: PICT-2018-1501 and PICT-2021-I-A-00240 to PW and PICT-2021-GRF-TII00418 to MM. SSF and SPP were supported with fellowships of the Consejo Nacional de Investigaciones Científicas y Técnicas.

## Additional information

### Competing interests

Pablo Wappner: Reviewing editor, *eLife*. The other authors declare that no competing interests exist.

### Funding

| Funder | Grant reference number | Author |
|---|---|---|
| Fondo para la Investigación Científica y Tecnológica | PICT-2021-I-A-00240 | Pablo Wappner |
| Fondo para la Investigación Científica y Tecnológica | PICT-2021-GRF-TII00418 | Mariana Melani |
| Fondo para la Investigación Científica y Tecnológica | PICT-2015-0372 | Pablo Wappner |
| Fondo para la Investigación Científica y Tecnológica | PICT-2017-1356 | Pablo Wappner |
| Fondo para la Investigación Científica y Tecnológica | PICT-2018-1501 | Pablo Wappner |

The funders had no role in study design, data collection, and interpretation, or the decision to submit the work for publication.

### Author contributions

Sofía Suárez Freire, Conceptualization, Investigation, Visualization, Methodology, Writing - original draft, Writing - review and editing; Sebastián Perez-Pandolfo, Sabrina Micaela Fresco, Julián Valinoti, Eleonora Sorianello, Methodology; Pablo Wappner, Resources, Supervision, Funding acquisition,

Project administration, Writing - review and editing; Mariana Melani, Conceptualization, Supervision, Funding acquisition, Investigation, Methodology, Writing - original draft, Writing - review and editing

### Author ORCIDs
Sofía Suárez Freire  http://orcid.org/0009-0003-2908-3110
Pablo Wappner  https://orcid.org/0000-0003-1517-0742
Mariana Melani  http://orcid.org/0000-0002-9491-932X

Reviewer #1 (Public review): https://doi.org/10.7554/eLife.92404.3.sa1
Reviewer #2 (Public review): https://doi.org/10.7554/eLife.92404.3.sa2
Reviewer #3 (Public review): https://doi.org/10.7554/eLife.92404.3.sa3
Author response https://doi.org/10.7554/eLife.92404.3.sa4

## Additional files

### Supplementary files
• MDAR checklist

### Data availability
All data generated or analyzed during this study are included in the manuscript and supporting files.

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
