## [Editor Report · eLife Assessment]

This study makes an **important** contribution by characterizing the role of the exocyst in secretory granule exocytosis in the *Drosophila* larval salivary gland. The results are **solid** and lead to the novel interpretation that the exocyst participates not only in exocytosis, but also in earlier steps of secretory granule biogenesis and maturation. However, the authors are urged to provide additional proof that the exocyst subunit knockdowns were effective and to acknowledge the possibility that inactivation of an essential exocytosis component could indirectly affect other parts of the secretory pathway.

---

## [Referee Report · Reviewer #1 (Public review)]

Suarez-Freire et al. analyzed here the function of the exocyst complex in the secretion of the glue proteins by the salivary glands of the *Drosophila* larva. This is a widely used, genetically accessible system in which the formation, maturation and precisely timed exocytosis of the glue secretory granules can be beautifully imaged. Using RNAi, the authors show that all units of the exocyst complex are required for exocytosis. They show that not just granule fusion with the plasma membrane is affected (canonical role), but also, with different penetrance, that glue protein is retained in the ER, secretory granules fail to fuse homotypically or fail to acquire maturation features. The authors document these phenotypes and postulate specific roles for the exocyst in these additional processes to explain them: exocyst as a Golgi-Golgi, Golgi-granule or granule-granule tether.

Compared to the initial submission, this revised version of the study presents strengthened evidence for these novel roles. In particular, authors show juxta-Golgi localization of exocyst components and disruption of the trans-Golgi compartment upon exocyst loss. Additionally, the revised study contains controls indicating that glue secretion defects prior to plasma membrane exocytosis are not due to polarity loss or unspecific poor health of cells.

---

## [Referee Report · Reviewer #2 (Public review)]

The manuscript from Wappner and Melani labs claims a novel for the exocyst subunits in multiple aspects of secretory granule exocytosis. This an intriguing paper for it suggests multiple roles of the exocyst in granule maturation and fusion with roles at the ER/Golgi interface, TGN, granule homotypic fusion.

A key strength is the breadth of the assays and study of all 8 exocyst subunits in a powerful model system (fly larvae). But why do KD of different exocysts have different effects on presumed granule formation? Also it can be hard to disentangle direct vs. secondary effects, as much of the TGN seems to be altered in the KDs. The authors ascribe many of the results to the holocomplex, but there are major differences between the proteins -- this may be all related to the different levels of expression (as the authors propose), but only limited mRNA was examined.

Unresolved Comments:

(A) Explanation variability of exocyst KD on the appearance of MSG. What is remarkable is a highly variable effect of different subunit KD on the percentage of cells with MLS (Fig. 4C). Controls = 100 %, Exo70=~75% (at 19 deg), Sec3 = ~30%, Sec10 = 0%, Exo84 = 100% ... This is interesting for the functional exocyst is an octameric holocomples, thus why the huge subunit variability in the phenotypes? One explanation is that the levels of KD varied between the subunits. Another is that not all subunits have equivalent roles (as seen for instance in exocyst's roles in autophagy).

This should be addressed by quantification of the KD of the 8 different exocyst proteins (and or mRNA as only 2 subunits were studied). If their data holds up then the underlying mechanism here needs to be considered. (Note: there is some precedent from the autophagy field of differential exocyst effects).

(B) Golgi: It is unclear from their model (Fig. 5) why after exocyst KD of Sec15 the cis-Golgi is more preserved than the TGN, which appears as large vacuoles.

(C) Granule homotypic fusion. Over-expression of just one subunit, Sec15-GFP, made giant secretory granules (SG) that were over 8 microns big. Does it act like a seed to promote exocyst assembly as the authors propose? If so is there evidence that there is biochemically more holocomplex with expression of Sec15, but not other subunits?

(D) The authors should better frame their interpretations of other studies of the exocyst that includes role in autophagy, Palade body trafficking and differential roles of the subunits.

In summary, there clearly are striking new effects on secretory granule biogenesis by dysfunction of the exocyst which are important and should inspire other studies for new roles of the exocyst; e.g. in non cannonical roles. Secondly, the power of the system to partially deplete proteins (if further validated) suggests that one may need to consider protein expression as an important variable that can be used to unmask multiple phenotypes in granule maturation. Last this paper implies new roles of the exocyst in homotypic fusion, which could be investigated in future work.

---

## [Referee Report · Reviewer #3 (Public review)]

Freire and co-authors examine the role of the exocyst complex during the formation and secretion of mucins from secretory granules in the larval salivary gland of *Drosophila melanogaster*. Using transgenic lines with a tagged Sgs3 mucin, the authors KD expression of exocyst subunit members and observe a defect in secretory granules with a heterogeneity of phenotypes. By carefully controlling RNAi expression using a Gal4-based system, the authors can KD exocyst subunit expression to varying degrees. The authors find that the stronger the inhibition of expression of the exocyst is, the earlier the defect is in the secretory pathway. The manuscript is well written, the model system is physiological, and the techniques are innovative.

In my initial review, my major concern was the pleiotropic effect of the loss of exocyst. The authors have responded to this point with clarity and have argued that the multiple localisations of exocyst during the Sgs3 synthesis programme indicate it is likely a direct phenotype. They also performed some analysis of PM lipids but did not detect a difference. I accept the arguments presented. However, I remain concerned that these are due to a pleiotropic effect. It is very hard to absolutely prove a direct effect, and due to the unusual claim and nature of the evidence (depletion levels), I think that there is still the possibility of this being an indirect effect. Perhaps it is just worth the authors writing a paragraph in the discussion, at least accepting the possibility that it is an indirect effect so future readers are aware of that.

---

## [Author Response]

The following is the authors’ response to the original reviews.

**Public Reviews:**

**Reviewer 1:**
(1) General comment: The evidence for these highly novel, potentially interesting roles (of the exocyst) would need to be more compelling to support direct involvement.

We wish to thank the reviewer for his/her comments, and for considering that the proposed functions are highly novel and potentially interesting. To strengthen the evidence supporting the new roles of the exocyst, we have performed a number of additional experiments that are depicted in novel figures or figure panels of the new version of the manuscript. Particularly, we aimed at providing further support of the direct involvement of the exocyst in different steps of the regulated secretory pathway. Please see the details below.

(2) For instance, the localization of exocyst to Golgi or to granule-granule contact sites does not seem substantial.

We have performed quantitative colocalization studies, as suggested by the reviewer to further substantiate our initial findings. We have carefully analysed GFP-Sec15 distribution in relation to the Golgi complex and secretory Glue granules at relevant time points of salivary gland development. Overall, we found that GFP-Sec15 distribution is dynamic during salivary gland development. Before Glue synthesis (72 h AEL), Sec15 was observed in close association (defined as a distance equal to, or less than 0.6 µm) with the Golgi complex (please see below Author response image 1). This association was lost once Glue granules have begun to form (96 h AEL). Importantly, we do not see relevant association between GFP-Sec15 and the ER (please see Author response image 2). These observations support our conclusion that the exocyst plays a role at the Golgi complex. New images supporting these conclusions, as well as quantitative data, have been included in Figure 5 of the new version of the manuscript. In addition, real time imaging, as well as 3D reconstruction analyses, confirming the close association between Sec15 and Golgi cisternae are now included in the manuscript. Please see Supplementary Videos 1-3. These new data are described in the text lines 200-210 of the Results section and text lines 359368 of the Discussion section.

Interestingly, at the time when Sec15-Golgi association is lost (96 h AEL), Sec15 foci associate instead with newly formed secretory granules (< 1µm diameter). This association persists during secretory granule maturation (100-116 h AEL), when Sec15 foci localize specifically in between neighbouring, immature secretory granules. When maturation has ended and Glue granule exocytosis begins (116-120 h AEL), this localization between granules is lost. These observations are consistent with a role of the exocyst in homotypic fusion during SG maturation. We have included new images showing that association between Sec15 and secretory granules is dynamic and depends on the developmental stage. We have quantified this association both during maturation and at a stage when SGs are already mature. We have in addition performed a 3D reconstruction analysis of these images to confirm the close association between Sec15 and immature SGs. These new data are now depicted in Figure 7BC, Supplementary Videos 4-5, and described in text lines 216-221 of the Results section. In addition, a lower magnification image is provided below in this letter (Author response image 3), quantifying the proportion of Sec15 foci localized in between SGs (yellow arrows) relative to the total number of Sec15 foci (yellow arrows + green arrowheads).

**Author response image 1. sa4fig1:** Criteria utilized to define Sec15 focithat were“associated” or“not associated” withthe trans-Golgi network in the experiments of Figure 5C-E of the manuscript. When the distance between maximal intensities of GFP-Sec15 and Golgi-RFP signals was equal or less than 0.6 µm, the signals were considered “associated” (upper panels). When the distance was more than 0.6 µm, the signals were considered “not associated” (lower panels).

**Author response image 2. sa4fig2:** Criteria utilized to define Sec15 focithat were“associated” or“not associated” withthe ERin the experiments of Figure 5A-Bof the manuscript. When the distance between maximal intensities of GFP-Sec15 and KDEL-RFP signals was equal or less than 0.6 µm, the signals were considered “associated”. When the distance was more than 0.6 µm, the signals were considered “not associated”.

**Author response image 3. sa4fig3:** The exocyst complex associates with immature SGs but not with mature SGs. (A) GFP-Sec15 foci (cyan) and SGs (red) are shown in cells bearing Immature SGs or (B) with mature SGs. Yellow arrows indicate GFP-Sec15 foci localized in between SGs; green arrowheads indicate GFP-Sec15 foci that arenot in between SGs. (C) Quantification of the percentage (%) of Sec15 foci localized in between SGs respect to the total number of Sec15 foci in cells filled with immature SGs (ISG)vs cells with mature SGs (MSG).

It is interesting to mention that previous evidence from mammalian cultured cells (Yeaman et al, 2001) show that the exocyst localizes both at the trans-Golgi network and at the plasma membrane, weighing in favour of our claim that the exocyst is required at various steps of the exocytic pathway. Thus, the exocyst may play multiple roles in the secretion pathway in other biological models as well. This concept has now been included at the Discussion section of the revised version of the manuscript (lines 359-368).

To make the conclusions of our work clearer, in the revised version of the manuscript, we have now included a graphical abstract, summarizing the dynamic localization of the exocyst in relation to the processes of SG biogenesis, maturation and exocytosis reported in our work.

(3) Instead, it is possible that defects in Golgi traffic and granule homotypic fusion are not due to direct involvement of the exocyst in these processes, but secondary to a defect in canonical exocyst roles at the plasma membrane. A block in the last step of glue exocytosis could perhaps propagate backward in the secretory pathway to disrupt Golgi complexes or cause poor cellular health due to loss of cell polarity or autophagy.

We thank the reviewer for these thoughtful comments. We have performed a number of additional experiments to assess “cellular health” or to identify possible defects in cell polarity after knock-down of exocyst subunits. These new data have been included in new supplementary figures 5 and 6 of the revised version of the manuscript (please see below).

In our view, the precise localization of GFP-Sec15 at the Golgi complex (Figure 5C-E), as well as in between immature secretory granules (Figure 7B-D), argues in favour of a direct involvement of the exocyst in SG biogenesis and homofusion respectively.

We truly appreciate the comment of the reviewer raising the possibility that the defects that we observe at early steps of the pathway (SG biogenesis and SG maturation) may actually stem from a backward effect of the role of the exocyst in SG-plasma membrane tethering. We wish to respectfully point out that the processes of biogenesis, maturation and plasma membrane tethering/fusion of SGs do not occur simultaneously in the *Drosophila* larval salivary gland *in vivo*, as they do in other secretory model systems (i.e. cell culture). In this regard, the experimental model is unique in terms of synchronization. In each cell of the salivary gland, the three processes (biogenesis, maturation and exocytosis) occur sequentially, and controlled by developmental cues. At the developmental stage when SGs fuse with the plasma membrane, SG biogenesis has already ceased many hours earlier: SG biogenesis occurs at 96-100 hours after egg lay (AEL), SG maturation takes place at 100-112 hours AEL, and SG-plasma membrane fusion happens only when all SGs have undergone maturation and are ready to fuse with the plasma membrane at 116-120 h AEL. Thus, in our view it is not conceivable that a defect in SG-plasma membrane tethering/fusion (116-120 h AEL) may affect backwards the processes of SG biogenesis or SG maturation, which have occurred earlier in development (96-112 h AEL).

As suggested by the reviewer, we have analysed several markers of cellular health and cell polarity, comparing conditions of exocyst subunit silencing (exo70RNAi, sec3RNAi or exo84RNAi) with wild type controls (whiteRNAi). These new data are depicted in Supplementary Figures 5 and 6, and described in lines 172-179 of the Results section of the revised version of the manuscript. Noteworthy, for these experiments we have applied silencing conditions that block secretory granule maturation, bringing about mostly immature SGs. Our analyses included: (1) Subcellular distribution of PI(4,5)P2, (2) subcellular distribution of the tetraspanin CD63, (3) of Rab11, (4) of filamentous actin, and (5) of CD8. We have also compared (6) nuclear size and nuclear general morphology, (7) the number and distribution of mitochondria, (8) morphology and subcellular distribution of the cis- and (9) trans-Golgi networks. Finally, (10) we have compared basal autophagy in salivary cells with or without knocking down exocyst subunits. The markers that we have analysed behaved similarly to those of control salivary glands, suggesting that the observed defects in regulated exocytosis indeed reflect different roles of the exocyst in the secretory pathway, rather than poor cellular health or impaired cell polarity.

Our conclusions are in line with previous studies in which apico-basal polarity, Golgi complex morphology and distribution, as well as apical membrane trafficking were also evaluated in exocyst mutant backgrounds, finding no anomalies (Jafar-Nejad et al, 2005).

Conversely, in studies in which apical polarity was disturbed by interfering with Crumbs levels, SG biogenesis, maturation and exocytosis were not affected (Lattner et al, 2019), indicating that these processes not necessarily interfere with one another.

(4) Final recommendation: In the absence of stronger evidence for these other exocyst roles, I would suggest focusing the study on the canonical role (interesting, as it was previously reported that *Drosophila* exocyst had no function in the salivary gland and limited function elsewhere [DOI: 10.1034/j.1600-0854.2002.31206.x]), and leave the alternative roles for discussion and deeper study in the future.

We appreciate the reviewer´s recommendation. However, we believe that the major strength of our work is the discovery of non-canonical roles of the exocyst complex, unrelated to its function as a tethering complex for vesicle-plasma membrane fusion. We believe that in the new version of our manuscript, we provide stronger evidence supporting the two novel roles of the exocyst:

a) Its participation in maintaining the normal structure of the Golgi complex, and b) Its function in secretory granule maturation.

**Reviewer 2:**
(5) General comment: A key strength is the breadth of the assays and study of all 8 exocyst subunits in a powerful model system (fly larvae). Many of the assays are quantitated and roles of the exocyst in early phases of granule biogenesis have not been ascribed.

We are grateful that the reviewer appreciates the novelty of our contribution.

(6) However there are several weaknesses, both in terms of experimental controls, concrete statements about the granules (better resolution), and making a clear conceptual framework. Namely, why do KD of different exocysts have different effects on presumed granule formation

The reviewer has raised a point that is central to the interpretation of all our data throughout the manuscript. The short answer is that the extent of RNAi-dependent silencing of exocyst subunits determines the phenotype:

(1) Maximum silencing affects Golgi complex morphology and prevents SG biogenesis. (2) Intermediate silencing blocks SG maturation, without affecting Golgi complex morphology and SG biogenesis. (3) Weak silencing blocks SG tethering and fusion with the plasma membrane, without affecting Golgi complex morphology, SG biogenesis or SG maturation.

In other words, (1) Low levels of exocyst subunits are sufficient for normal Golgi complex morphology and SG biogenesis. (2) Intermediate levels of exocyst subunits are sufficient for SG maturation (and also sufficient for SG biogenesis). (3) High levels of exocyst subunits are required for SG tethering and subsequent fusion with the plasma membrane.

Based on the above notion, we have exploited the fact that temperature can fine-tune the level of Gal4/UAS-dependent transcription, thereby achieving different levels of silencing, as shown by Norbert Perrimon et al in their seminal paper “the level of RNAi knockdown can also be altered by using Gal4 lines of various strengths, rearing flies at different temperatures, or via coexpression of UAS-Dicer2” (Perkins et al, 2015).

We found in our system that indeed, by applying appropriate silencing conditions (RNAi line and temperature) to any of the eight subunits of the exocyst, we have been able to obtain one of the three alternative phenotypes: Impaired SG biogenesis, or impaired SG maturation, or impaired SG tethering/fusion with the plasma membrane.

These concepts are summarized below in Author response image 4. Please see also at point 26, the general comment of Reviewer #3.

We have conducted qRT-PCR assays to provide experimental support to the notions summarized above in Author response image 4. We measured the remaining levels of mRNAs of some of the exocyst subunits, after inducing RNAi-mediated silencing at different temperatures, or with different RNAi transgenic lines. The remaining RNA levels after silencing correlate well with the observed phenotypes, following the predictions of Author response image 4 and summarized in Author response image 5. These new data are now shown in Supplementary Figure 2 of the revised version of the manuscript, and described in lines 153-159 at the Results section.

(7) Why does just overexpression of a single subunit (Sec15) induce granule fusion?

The reviewer raises a very important point. Based on available data from the literature, Sec15 behaves as a seed for assembly of the holocomplex and it also mediates the recruitment of the holocomplex to SGs through its interaction with Rab11 (Escrevente et al, 2021; Bhuin and Roy, 2019; Wu et al, 2005; Zhang et al, 2004; Guo et al, 1999). Thus, overexpression of Sec15 is expected to enhance exocyst assembly, thereby potentiating the activities carried out by the complex in the cell, including SG homofusion. In the revised version of the manuscript we have also performed the overexpression of Sec8, finding that, unlike Sec15, Sec8 fails to induce homotypic fusion. These results were expected, as they confirm that Sec8 does not behave as a seed for mounting the whole complex. These new data have been included in Figure 7E-H, and are described in text lines 221-229 of the Results section.

**Author response image 4. sa4fig4:** Conceptual model of RNAi expression at different temperatures , remaining levels of mRNA/protein levels and phenotypes obtained at each temperature.

**Author response image 5. sa4fig5:** qRT-PCR assays presented in Supplementary Figure 2 are shown in combination with the phenotypes observed at each of the conditions analyzed. Note the correlation between phenotypes and the extent of mRNA downregulation.

(8) While the paper is fascinating, the major comments need to be addressed to really be able to make better sense of this work, which at present is hard to disentangle direct vs. secondary effects, especially as much of the TGN seems to be altered in the KDs.

We hope that our response to point (6) has helped to clarify this important point raised by the Reviewer. After applying silencing conditions where normal structure of the trans-Golgi network is impaired, SG biogenesis does not occur. Thus, since SGs do not form, it is not conceivable to detect defects in SG maturation or SG fusion with the plasma membrane in the same cell.

(9) The authors conveniently ascribe many of the results to the holocomplex, but their own data (Fig. 4 and Fig. 6) are at odds with this.

This is another central point of our work, so we thank the reviewer for his/her comment. In Figures 4A, 7A and 9A of the revised version of the manuscript, we show that, by inducing appropriate levels of silencing of any of the 8 subunits of the exocyst, each of the three alternative phenotypic manifestations can occur. In our opinion, this argues in favour of a function for the whole exocyst complex in each of the three specific activities proposed in our study: (1) SG biogenesis, (2) SG maturation, and (3) SG tethering/fusion with the plasma membrane. In detailed characterizations of these three phenotypes performed throughout the study, we decided to induce silencing of just two or three of the subunits of the exocyst, assuming that the whole complex accounts the mechanisms involved.

Major comments(10) Resolution not sufficient. Identification of "mature secretory granules" (MSG) in Fig. 3 is based on low-resolution images in which the MSG are not clearly seen (see control in Fig. 3A) and rather appear as a diffuse haze, and not as clear granules. There may be granules here, but as shown it is not clear. Thus it would be helpful to acquire images at higher resolution (at the diffraction limit, or higher) to see and count the MSG.

We thank the reviewer for raising this point, as it may not be straightforward to the reader to identify the SGs throughout the figures of our study. To make it clearer, in Figure 3A (magnified insets on the right), we have delimitated individual SGs with a green dotted line, and included diagrams (far right), which we hope will help the identification of SGs. In Figure 3B, we show that after silencing Sec84, a mosaic phenotype was observed: In some cells SGs fail to undergo maturation, and remain smaller than normal. In other cells of this mosaic phenotype, biogenesis of SGs was impaired and the fluorescent cargo remained trapped in a mesh-like structure (that we later show that corresponds to the ER). The dotted line marks individual SGs, and the diagrams included on the right intend to help the interpretation of the phenotype. The mesh-like structures where Sgs3-GFP was retained are also marked with dotted line, and schematized on the right. These new schemes are described in the Figure 3 caption of the revised version of the manuscript.

We wish to mention that all the confocal images depicted in this figure and throughout the manuscript have been captured at high resolution, with a theoretical resolution limit of 168177nm (d = γ/2NA). Given that secretory granules range from 0.8-7µm in diameter, the resolution is more than sufficient to clearly resolve these structures.

(11) Note: the authors are not clear on which objective was used. Maybe the air objective as the resolution appears poor.

In this particular figure, we have utilized a Plan-Apochromat 63X/1.4NA oil objective of the inverted Carl Zeiss LSM 880 confocal microscope (mentioned in materials and methods).

(12) They need to prove that the diffuse Sgs3-GFP haze is indeed due to MSG.

If we interpret correctly the concern of the reviewer, what he/she calls “diffuse haze” is actually the distribution of Sgs3-GFP within individual SGs, which, as previously reported by other authors, is not homogeneous at this stage (Syed et al. 2022). We hope that the diagrams that we have included in Figure 3 A, B (point 10) will help the readers interpreting the images.

(13) Related it is unclear what are the granule structures that correspond to Immature secretory granules (ISG) and cells with mesh-like structures (MLS)?

We are confident that the diagrams now included in Figure 3A and B will help the interpretation, and particularly to identify immature granules and the mesh-like structure generated after silencing of exocyst subunits.

(14) Similarly, Sgs3 images of KD of 8 exocyst subunits were interpreted to be identical, in Fig. 4, but the resolution is poor.

We hope that the issue related to resolution of our images has been properly addressed in the response to point (10) of this letter. In Figure 4A, we show that after silencing of any of the 8 subunits (with the appropriate conditions), in all cases SG biogenesis was impaired, and Sgs3GFP was instead retained in a mesh-like structure. Images obtained after silencing different exocyst subunits are of course not identical, but in all cases, a mesh-like structure has replaced the formation of SGs (Figure 4A). Hopefully, the diagrams now included in Figure 3A and B help the correct interpretation of the phenotypes throughout the study.

To demonstrate that the structure in which Sgs3-GFP was retained upon exocyst complex knockdown corresponds to the ER, we performed a colocalization analysis between Sgs3-GFP and the ER markers GFP-KDEL or Bip-sfGFP-HDEL, after which we calculated the Pearsons Coefficient, which indicated substantial colocalization (Figure 4B-G and Supplementary Figures 7 and 8). These new data are described in lines 196-199 of the revised version of the manuscript. To facilitate the visualization of the results, in the revised version of the manuscript we have included magnified cropped areas of the images shown in Figure 4A.

(15) What is remarkable is a highly variable effect of different subunit KD on the percentage of cells with MLS (Fig. 4C). Controls = 100 %, Exo70=~75% (at 19 deg), Sec3 = ~30%, Sec10 = 0%, Exo84 = 100% ... This is interesting for the functional exocyst is an octameric holocomples, thus why the huge subunit variability in the phenotypes? The trivial explanation is either: (i) variable exocyst subunit KD (not shown) or (ii) variability between experiments (no error bars are shown). Both should be addressed by quantification of the KD of different proteins and secondly by replicating the experiments.

We agree with the reviewer statement. We believe that both, variability of KD efficiency (i) and variability between experiments (ii) contribute to the variable effect observed after knocking down the different subunits. As detailed in the response to point (6), we have performed qRT-PCR determinations to confirm that the severity of the phenotype depends on the efficiency of RNAimediated silencing. We chose to analyse in detail the effect on the subunits exo70 and sec3, which were those with the highest phenotypic differences between the three silencing temperatures utilized. We found that as expected, the levels of silencing were temperaturedependent, being higher at 29°C and lower at 19°C. These data were included in Supplementary Figure 2, and described lines 153-159 of the Results section and also summarized in Author response images 4 and 5 of this rebuttal letter.

We thank the reviewer for his/her comment on the replication of experiments and statistics. We failed to include detailed numerical information in the original submission, such as the number of replicas and standard deviations of the data depicted in Figure 3C and Supplementary Figure 1, so we apologize for this omission. In the revised version of the manuscript, we have included a table (Supplementary Table 3) in which all the raw data of Figure 3C and Supplementary Figure 1, including standard deviations, are now depicted.

(16) If their data holds up then the underlying mechanism here needs to be considered.(Note: there is some precedent from the autophagy field of differential exocyst effects)

Our proposed mechanism is essentially that the holocomplex is required for multiple processes along the secretory pathway. Each of these actions (Golgi structure maintenance, SG maturation and SG tethering/fusion with the plasma membrane) requires different amounts of holocomplex activity, being this the reason why each phenotype manifests at different levels of RNAi-mediated silencing (Author response image 4 of this letter). The model predicts that Golgi structure maintenance requires minimal levels of complex activity, and that is why strong knock-down of exocyst subunits is required to obtain this phenotype. In line with our results, it has been reported that other tethering complexes of the CATCHR family are also required for maintaining Golgi cisternae stuck together (D'Souza et al, 2020; Khakurel and Lupashin, 2023; Liu et al, 2019). One possibility is that the exocyst may play a redundant role in the maintenance of the normal structure of the Golgi complex, along with other CATCHR complexes. This potential redundancy could explain why severe exocyst knock-down is required to observe structural anomalies at this organelle. On the other end of the spectrum, we propose that tethering/fusion with the plasma membrane is very susceptible to even slight reduction of complex activity, so that mild RNAi-mediated silencing is sufficient to provoke defects in this process. This proposed model is depicted in Author response image 4 and discussed in lines 395-405 of the Discussion section.

(17) In the salivary glands the authors state that the exocyst is needed for Sgs3-GFP exit from the ER. First, Pearson's coefficient should be shown so as to quantitate the degree of ER localizations of all KDs.

We thank the reviewer for this comment that helped us to strengthen the observation that when SG biogenesis is impaired, Sgs3-GFP remains trapped in the ER. In the revised version of the manuscript, we have calculated Pearson´s coefficient to assess colocalization between ER markers (GFP-KDEL or Bip-sfGFP-HDEL) and Sgs3-GFP in salivary gland cells that express sec15RNAi. The Pearson’s coefficient was around 0.6 for both ER markers, indicating that colocalization with Sgs3-GFP was substantial (Supplementary Figure 8, text lines 196-199 of the Results section).

(18) Second, there should be some rescue performed (if possible) to support specificity.

As suggested by the reviewer, we have performed a rescue experiment of the phenotype provoked by the expression of sec15 RNAi, which consisted on the retention of Sgs3-GFP in the endoplasmic reticulum: Expression of Sec15-GFP reverted substantially the ER retention phenotype, rescuing SG biogenesis and also SG maturation in most cells (over 60% of the cells). These new data are now shown in Supplementary Figure 4, and described in lines 168-171 of the Results section.

(19) Third, importantly other proteins that should traffic to the PM need to be shown to traffic normally so as to rule out a non-specific effect.

We have addressed this issue (also mentioned by Reviewer #1), by analyzing the localization of a number of polarization markers, finding that the overall polarization of the cell was not affected by loss of function of exocyst subunits. Please, see our response to the point (3) raised by Reviewer #1. The new data showing cell polarization markers are shown in Supplementary Figure 6 of the revised version of the manuscript, and described on text lines 172-179 of the Results section.

(20) It is unclear from their model (Fig. 5) why after exocyst KD of Sec15 the cis-Golgi is more preserved than the TGN, which appears as large vacuoles. This is not quantitated and not shown for the 8 subunits.

We thank the reviewer for this relevant comment. We agree that the phenotype of either, sec15 or sec3 loss-of-function cells manifests differently with cis-Golgi and trans-Golgi markers. While the cis-Golgi marker looked fragmented and aggregated, the trans-Golgi marker adopted a swollen appearance. However, in our view, the different appearance of the two markers does not necessarily imply that one compartment is more preserved than the other. In the revised version of the manuscript, we have quantified the penetrance of the phenotypes provoked by sec15 or sec3 silencing, using both cis-Golgi and trans-Golgi markers. In both cases, the penetrance was high, although even higher with the trans-Golgi marker. These new data are now depicted in Supplementary Figure 9 of the revised version of the manuscript.

It is interesting to mention that in HeLa cells, as well as in the retinal epithelial cell line hTERT, Golgi phenotypes similar to those we have described here have been reported after loss-offunction of other tethering complexes, which were shown to maintain the Golgi cisternae stuck together, including the GOC and GARP complexes (D'Souza et al, 2020, Khakurel and Lupashin, 2023; Shijie Liu et al, 2019). As we did throughout our work, not every aspect of the analysis included the silencing of all eight subunits. In this case, we chose to silence Sec3 and Sec15. Please note that we have modified the model depicted in Figure 6E-F, to highlight the cis- and transGolgi phenotypes upon exocyst knock-down, as well as the localization of the exocyst in cisternae of the Golgi complex.

(21) Acute/Chronic control: It would be nice to acutely block the exocyst so as to better distinguish if the effects observed are primary or secondary effects (e.g. on a recycling pathway).

We thank the reviewer for raising this important issue. To address this point, and to be able to induce silencing of exocyst subunits at specific time intervals of larval development, we utilized a strategy based on a thermosensitive variant of the Gal4 inhibitor Gal80 (Gal80ts) (Lee and Luo, 1999). We blocked Gal4 activity (and therefore RNAi expression) by maintaining the larvae at 18 °C during the 1st and 2nd instars (until 120 hours after egg lay), and then induced the activity of Gal4 specifically at the 3rd larval instar by raising the temperature to 29 ºC, a condition in which Gal80ts becomes inactive. After silencing the expression of sec3 or sec15 at the 3rd larval instar only, the phenotype was very similar to that observed after chronic silencing of exocyst subunits (larvae maintained at 29 ºC all throughout development, where Gal4 was never inhibited). These observations suggest that the defects observed in the secretory pathway after knock down of exocyst subunits reflect genuine functions of the exocyst in this pathway, rather than a secondary effect derived from impaired development of the salivary glands at early larval stages. These new results are now shown in Supplementary Figure 3, and described in manuscript lines 160-171 of the Results section.

(22) Granule homotypic fusion. Strangely over-expression of just one subunit, Sec15-GFP, made giant secretory granules (SG) that were over 8 microns big! Why is that, especially if normally the exocyst is normally a holocomplex. Was this an effect that was specific to Sec15 or all exocyst subunits? Is the Sec15 level rate limiting in these cells? It may be that a subcomplex of Sec15/10 plays earlier roles, but in any case this needs to be addressed across all (or many) of the exocyst subcomplex members.

Please, see our response to point (7) of this letter. Sec15 is believed to act as a seed for the formation of the whole complex.

(23) In summary, there are clearly striking effects on secretory granule biogenesis by dysfunction of the exocyst, however right now it is hard to disentangle effects on ER-Golgi traffic, loss of the TGN, and a problem in maturation or fusion of granules.

As discussed in detail in our response to the point 3 raised by Reviewer #1, the secretory pathway is highly synchronized in each of the cells of the *Drosophila* salivary gland. SG biogenesis, SG maturation and SG fusion with the plasma membrane never occur simultaneously in the same cell. Thus, in a cell in which ER-Golgi traffic is impaired (and SG biogenesis does not occur), SGs do not exist, and therefore, they cannot exhibit defects in the process of maturation or fusion with the plasma membrane. In summary, we believe that our work has shown that in *Drosophila* larval salivary glands the exocyst holocomplex is required for (at least) three functions along the secretory pathway: (1) To maintain the appropriate Golgi complex architecture, thus enabling ERGolgi transport; (2) For secretory granule maturation: both, homotypic fusion and acquisition of maturation factors; (3) For secretory granule exocytosis: secretory granule tethering to enable subsequent fusion with the plasma membrane. As mentioned above (point 6 of this letter), these three functions require different amounts of the holocomplex, and therefore can be revealed by inducing different levels of silencing.

(24) It is also confusing if the entire exocyst holocomplex or subcomplex plays a key role

The fact that, by silencing any of the subunits (with the appropriate conditions) it is possible obtain any of the 3 phenotypes (impaired SG biogenesis, impaired SG maturation or impaired SG fusion with the plasma membrane) argues in favour of a function of the complex as a whole in each of these three functions.

**Reviewer 3:**
(25) General comment: Freire and co-authors examine the role of the exocyst complex during the formation and secretion of mucins from secretory granules in the larval salivary gland of *Drosophila melanogaster*. Using transgenic lines with a tagged Sgs3 mucin the authors KD expression of exocyst subunit members and observe a defect in secretory granules with a heterogeneity of phenotypes. By carefully controlling RNAi expression using a Gal4-based system the authors can KD exocyst subunit expression to varying degrees. The authors find that the stronger the inhibition of expression of exocyst the earlier in the secretory pathway the defect. The manuscript is well written, the model system is physiological, and the techniques are innovative.

We appreciate the reviewer´s assessment of our work.

(26) My major concern is that the evidence underlying the fundamental claim of the manuscript that "the exocyst complex participates" in multiple secretory processes lacks direct evidence.

We thank the reviewer for raising this important issue. We believe that the analysis of Sec15 subcellular localization during salivary gland development (Figures 5, 7B-D and 9E-F), in combination with the detailed analysis of the phenotypes provoked by loss-of-function of each of the exocyst subunits, provide evidence supporting multiple functions of the exocyst in the secretory pathway. We have also included 3D reconstructions and videos of GFP-Sec15 colocalization with Golgi and SG markers to support exocyst localization associated to these structures (Supplementary Videos 1-7), text lines 200-210; 216-221 and 303-305.

(27) It is clear from multiple lines of evidence, which are discussed by the authors, that exocyst is essential for an array of exocytic events. The fundamental concern is that loss of homeostasis on the plasma membrane proteome and lipidome might have severe pleiotropic effects on the cell.

We agree with the reviewer that this is an important point that needed to be addressed. As discussed in detail above at the response to point 3 raised by Reviewer #1, we have analysed several plasma membrane markers (including a PI(4,5)P2 lipid reporter), and found that overall, plasma membrane integrity and polarity were not substantially affected (Supplementary Figure 6). In addition, we have analyzed several markers of general cellular “health” that indicate that salivary gland cells do not seem to be distressed by the reduction of exocyst complex activity (Supplementary Figure 5). These new data are described in lines 172-179 of the Results section.

(28) Perhaps the authors have more evidence that exocyst is important for homeotypic fusion of the SGs, as supported by the localisation of Sec15 on the fusion sites.

We believe that the fact that, by silencing any of the exocyst subunits (with the appropriate conditions), immature smaller-than-normal granules were observed, argus in favour that the exocyst as a whole participates in SG homofusion (Figure 7A). In addition, we have included more images, quantifications, 3D reconstructions and videos of GFP-Sec15 localized just at the contact sites between immature SGs. We have quantified and compared GFP-Sec15 localization at immature SG vs its localization at mature SGs, finding that localizes preferentially at immature SGs, supporting a role of the exocyst as a tethering complex during homotypic fusion (shown Figure 7B-C and Supplementary Videos 4-6, and described in lines 216-221 of the Results section). Please see also our response to the point 2 raised by reviewer 1 in this rebuttal letter, and to Author response image 3 above in this letter.

(29) The second question that I think is important to address is, what exactly do the varying RNAi levels correspond to in terms of experiments, and have these been validated? Due to the fundamental claim being that the severity of the phenotype being correlated with the level of KD, I think validation of this model is absolutely essential.

We thank the Reviewer for raising this important point, and agree it was lacking in the original version of our manuscript. As discussed in our response to the point (6) raised by Reviewer #2, we have performed qRT-PCR determinations for exo70 and sec3 mRNA levels after inducing silencing of these subunits at different temperatures, or with different RNAi transgenic lines. The remnant mRNA levels correlate well with the observed phenotypes. Please see Supplementary Figure 2 of the revised manuscript, and Author response image 5 of this rebuttal letter; described in lines 155-159 of the Results section.

**Recommendations for the authors:**

**Reviewer #1 (Recommendations For The Authors):**
- The authors assert in the discussion that exocyst involvement in constitutive secretion is well documented. This is based on a very recent study in mammalian culture cells. Therefore, I would not dismiss the issue as completely settled. Furthermore, a previous study of *Drosophila* sec10 reported no roles outside the ring gland (DOI: 10.1034/j.1600-0854.2002.31206.x).

We have included these observations in the Discussion section. Lines 326-329.

- A salivary gland screening by Julie Brill's lab reported exocyst components as hits (DOI: 10.1083/jcb.201808017).

We have referred to this paper in the Discussion section. Lines 326-329.

- It should be explained in more detail what is measured in graphs 7C, F, and others quantifying fluorescence around secretory granules. Looking at the images, the decrease in Rab1 and Rab11 seems less convincing.

We have made a clearer description of how fluorescence intensity was measured in the Methods section lines 558-561. Also, we have uploaded a source data file in which the raw data of each experiment used for quantifications are disclosed.

Please note that the data indicates that Rab11 levels are higher in sec5 (Figure 8J-L) and sec3 (supplementary Figure 11M-R).

**Reviewer #2 (Recommendations For The Authors):**
No major issues.Writing - The authors should better frame their interpretations of other studies of the exocyst that include the role in autophagy, Palade body trafficking, and differential roles of the subunits.

We have discussed these specific points in the Discussion section, lines 348-355 and 409-410.

Minor - Fig. 6A: Why are variable temperatures (19-29 deg C used for the 8 KD experiments)?Please show it all at the same temperature (control too).

The need for the usage of specific temperatures to obtain specific phenotypes with each of the RNAi lines used was explained in point 6 of this letter.

**Reviewer #3 (Recommendations For The Authors):**
In the abstract, the authors refer to the exocytic process and go on to describe secretory granule biogenesis and exocytosis. However, there are many exocytic processes aside from secretory granule biogenesis, and I think the authors should clarify this.

Corrected in the Abstract. Lines 19-21

Page 17 Thomas, 2021 reference, there is a glitch with the reference.

Thanks for noticing. Fixed.

References

Bhuin T, Roy JK. Developmental expression, co-localization and genetic interaction of exocyst component Sec15 with Rab11 during *Drosophila* development. Exp Cell Res. 2019 Aug 1;381(1):94-104. doi: 10.1016/j.yexcr.2019.04.038. Epub 2019 May 7. PMID: 31071318.

D'Souza Z, Taher FS, Lupashin VV. Golgi inCOGnito: From vesicle tethering to human disease. Biochim Biophys Acta Gen Subj. 2020 Nov;1864(11):129694. doi: 10.1016/j.bbagen.2020.129694. Epub 2020 Jul 27. PMID: 32730773; PMCID: PMC7384418.

Escrevente C, Bento-Lopes L, Ramalho JS, Barral DC. Rab11 is required for lysosome exocytosis through the interaction with Rab3a, Sec15 and GRAB. J Cell Sci. 2021 Jun 1;134(11):jcs246694. doi: 10.1242/jcs.246694. Epub 2021 Jun 8. PMID: 34100549; PMCID: PMC8214760.

Guo W, Roth D, Walch-Solimena C, Novick P. The exocyst is an effector for Sec4p, targeting secretory vesicles to sites of exocytosis. EMBO J. 1999 Feb 15;18(4):1071-80. doi: 10.1093/emboj/18.4.1071. PMID: 10022848; PMCID: PMC1171198.

Jafar-Nejad H, Andrews HK, Acar M, Bayat V, Wirtz-Peitz F, Mehta SQ, Knoblich JA, Bellen HJ. Sec15, a component of the exocyst, promotes notch signaling during the asymmetric division of *Drosophila* sensory organ precursors. Dev Cell. 2005 Sep;9(3):351-63. doi: 10.1016/j.devcel.2005.06.010. PMID: 16137928.

Khakurel A, Lupashin VV. Role of GARP Vesicle Tethering Complex in Golgi Physiology. Int J Mol Sci. 2023 Mar 23;24(7):6069. doi: 10.3390/ijms24076069. PMID: 37047041; PMCID: PMC10094427.

Lattner J, Leng W, Knust E, Brankatschk M, Flores-Benitez D. Crumbs organizes the transport machinery by regulating apical levels of PI(4,5)P2 in *Drosophila*. Elife. 2019 Nov 7;8:e50900. doi: 10.7554/eLife.50900. PMID: 31697234; PMCID: PMC6881148.

Lee T, Luo L. Mosaic analysis with a repressible cell marker for studies of gene function in neuronal morphogenesis. Neuron. 1999 Mar;22(3):451-61. doi: 10.1016/s08966273(00)80701-1. PMID: 10197526.

Liu S, Majeed W, Grigaitis P, Betts MJ, Climer LK, Starkuviene V, Storrie B. Epistatic Analysis of the Contribution of Rabs and Kifs to CATCHR Family Dependent Golgi Organization. Front Cell Dev Biol. 2019 Aug 2;7:126. doi: 10.3389/fcell.2019.00126. PMID: 31428608; PMCID: PMC6687757.

Perkins LA, Holderbaum L, Tao R, Hu Y, Sopko R, McCall K, Yang-Zhou D, Flockhart I, Binari R, Shim HS, Miller A, Housden A, Foos M, Randkelv S, Kelley C, Namgyal P, Villalta C, Liu LP, Jiang X, Huan-Huan Q, Wang X, Fujiyama A, Toyoda A, Ayers K, Blum A, Czech B, Neumuller R, Yan D, Cavallaro A, Hibbard K, Hall D, Cooley L, Hannon GJ, Lehmann R, Parks A, Mohr SE, Ueda R, Kondo S, Ni JQ, Perrimon N. The Transgenic RNAi Project at Harvard Medical School: Resources and Validation. Genetics. 2015 Nov;201(3):843-52. doi: 10.1534/genetics.115.180208. Epub 2015 Aug 28. PMID: 26320097; PMCID: PMC4649654.

Wu S, Mehta SQ, Pichaud F, Bellen HJ, Quiocho FA. Sec15 interacts with Rab11 via a novel domain and affects Rab11 localization in vivo. Nat Struct Mol Biol. 2005 Oct;12(10):879-85. doi: 10.1038/nsmb987. Epub 2005 Sep 11. PMID: 16155582.

Yeaman C, Grindstaff KK, Wright JR, Nelson WJ. Sec6/8 complexes on trans-Golgi network and plasma membrane regulate late stages of exocytosis in mammalian cells. J Cell Biol. 2001 Nov 12;155(4):593-604. doi: 10.1083/jcb.200107088. Epub 2001 Nov 5. PMID: 11696560; PMCID: PMC2198873.

Zhang XM, Ellis S, Sriratana A, Mitchell CA, Rowe T. Sec15 is an effector for the Rab11 GTPase in mammalian cells. J Biol Chem. 2004 Oct 8;279(41):43027-34. doi: 10.1074/jbc.M402264200. Epub 2004 Jul 29. PMID: 15292201.